# Increasingly efficient chromatin binding of cohesin and CTCF supports chromatin architecture formation during zebrafish embryogenesis

Jonas Coßmann[1,4], Pavel I. Kos [2,5], Vassiliki Varamogianni-Mamatsi[3], Devin S. Assenheimer[1,4], Tobias A. Bischof [1,4], Timo Kuhn[1], Thomas Vomhof [1], Argyris Papantonis [3], Luca Giorgetti [2] & J. Christof M. Gebhardt [1,4] ✉

The three-dimensional folding of chromosomes is essential for nuclear functions such as DNA replication and gene regulation. The emergence of chromatin architecture is thus an important process during embryogenesis. To shed light on the molecular and kinetic underpinnings of chromatin architecture formation, we characterized biophysical properties of cohesin and CTCF binding to chromatin and their changes upon cofactor depletion using single-molecule imaging in live developing zebrafish embryos. We found that chromatin-bound fractions of both cohesin and CTCF increased significantly between the 1000-cell and shield stages, which we could explain through changes in both their association and dissociation rates. Moreover, increasing binding of cohesin restricted chromatin motion, potentially via loop extrusion, and showed distinct stage-dependent nuclear distribution. Polymer simulations with experimentally derived parameters recapitulated the experimentally observed gradual emergence of chromatin architecture. Our findings reveal molecular kinetics underlying chromatin architecture formation during zebrafish embryogenesis.

The three-dimensional organization of chromatin is important for nuclear functions such as DNA replication and gene regulation. In interphase, the organizational structures span several hierarchical levels, from chromosome territories[1] to compartments[2,3], topologically associating domains (TADs)[4–6], and DNA loops[6,7] to clutched and separated nucleosomes[8,9]. TADs and loops contribute to gene regulation by bringing together or insulating enhancers and promoters. They are formed by the architectural protein cohesin through loop extrusion and barrier-forming proteins such as the CCCTC-binding factor (CTCF)[10–13]. Moreover, concomitantly with replication, cohesin ensures the cohesion of sister chromatids[14,15].

Cohesin is a multi-subunit protein complex that forms a ring-like core complex through the ATPases Smc1/3 and Rad21[16]. Its ATPase activity and, thereby, the loop extrusion function of the core complex depend on Mau2-Nibpl[17–20], while Wapl determines the release of cohesin from DNA[21,22]. For the cohesion function, a fraction of cohesin is stabilized on chromatin through Sororin (Cdca5), which antagonizes the activity of Wapl[23–26]. The exact extrusion mechanism of cohesin-mediated loop extrusion is still unclear. Pieces of evidence point towards loading of the cohesin complex on the DNA, followed by cohesin reeling in DNA symmetrically or by periodically switching between one-sided extrusion[27–29]. CTCF binds in an orientation-

[1]Institute of Biophysics, Ulm University, Ulm, Germany. [2]Friedrich Miescher Institute for Biomedical Research, Basel, Switzerland. [3]Institute of Pathology, University Medical Center Göttingen, Göttingen, Germany. [4]Present address: Institute of Experimental Physics and IQST, Ulm University, Ulm, Germany. [5]Present address: Department of Biomedicine, University of Basel, Basel, Switzerland. ✉e-mail: christof.gebhardt@uni-ulm.de

dependent manner to specific DNA motifs, often forming a TAD boundary[6,30,31], and guides the formation of regulatory DNA loops by stalling the loop-extrusion process upon direct interaction with cohesin[32–34].

Chromatin architecture in interphase is absent in zygotes of animals as diverse as flies, zebrafish, and humans[35], but is gradually reestablished during early embryo development[36–42]. In Zebrafish embryos, the new formation of chromatin architecture starts around the 1000-cell stage, concurrent with zygotic genome activation (ZGA), when the embryo begins transcribing its own genome and becomes independent of maternal RNA and protein[40].

While important insight on the mechanistic underpinnings of cohesin-mediated loop extrusion and CTCF, including the interaction kinetics with chromatin and the cellular distribution have been obtained in vitro and in cells[17,18,34,43–51], detailed information on the temporal and spatial behavior in vivo is missing. In particular, it is unclear how frequently and for how long cohesin and CTCF interact with chromatin, how cohesin action affects chromatin motion, and how these parameters support the formation of chromatin architecture during embryo development.

Here, we characterized the binding of cohesin and CTCF to chromatin during early embryogenesis using single molecule tracking in live developing zebrafish embryos and polymer simulations. We measured the fraction of cohesin and CTCF molecules associated with chromatin, their association and dissociation kinetics, and the motion of cohesin-bound chromatin at multiple developmental stages. Our experiments and simulations revealed mechanistic insight into these architectural proteins that inform the process of chromatin architecture formation in zebrafish embryogenesis.

## Results

### The fraction of chromatin-bound cohesin and CTCF increases during development

To be able to visualize cohesin and CTCF in zebrafish embryos, we fused the cohesin subunit Rad21 and CTCF to a HaloTag (HT) (Fig. 1a and Methods). We injected mRNA encoding for either fusion protein together with mRNA encoding for the nuclear membrane marker GFP-lap2b into 1-cell stage embryos[52] (Fig. 1b and Methods), which resulted in ectopic protein expression of ca. (11.4 ± 2.6)% for HT-Rad21 and (24.0 ± 2.7)% for HT-CTCF of endogenous and maternally[40,53,54] provided protein in shield-stage embryos (Supplementary Fig. 1). We confirmed the incorporation of HT-Rad21 into the cohesin complex through co-immunoprecipitation of the N-terminally interacting Smc3 protein[55] (Supplementary Fig. 2). We further confirmed chromatin association of HT-Rad21 and HT-CTCF by subcellular protein fractionation (Supplementary Fig. 3). The HT-rad21 and HT-ctcf injected embryos developed normally during our imaging period and up to 48 hpf (Supplementary Fig. 4 and 5). Prior to imaging, we labeled HT-tagged proteins with HT-dye JF549[56] and transferred them onto glass-bottom imaging dishes. Illumination of embryos with either 488 nm or 561 nm laser light in a reflected light-sheet microscope[52,57] enabled detection the nuclear membrane and the signal of individual HT-Rad21 or HT-CTCF molecules in the nucleus.

We characterized the overall mobility of HT-Rad21 and HT-CTCF in the developmental stages between the 64-cell stage and the shield stage comprising the period of ZGA. In every stage, both species showed mobile and immobile molecules (Fig. 1c, Supplementary Movie 1 and 2 and Methods). To measure the fraction of immobile molecules, we recorded continuous movies of single HT-tagged molecules with 11.7 ms frame rate (Fig. 1c). We analyzed movies using TrackIt[58], compiled cumulative histograms of jump distances of tracked molecules (Supplementary Figs. 6 and 7) and analyzed these using a model comprising three components of Brownian diffusion (Supplementary Fig. 8 and Methods)[59–63]. The first, slow diffusion component with fraction $A_1$ represents immobile molecules

presumably bound to chromatin that show apparent Brownian diffusion mainly due to the limited localization precision of diffraction-limited imaging and slow motion of the underlying chromatin. The two other diffusion components with fractions $A_2$ and $A_3$ are used to approximate the anomalous diffusion that has been observed for mobile molecules in the nucleus[59,64–66]. For HT-Rad21, the fraction of immobile, potentially chromatin-bound molecules increased ca. 8.7-fold, and for HT-CTCF 1.5-fold from 64-cell to shield stage embryos. In contrast, the intermediate and fast diffusion fractions, which presumably comprise freely diffusing molecules in the crowded nucleus, remained roughly constant (Fig. 1d, e). Thus, HT-Rad21 and HT-CTCF are more and more driven to the chromatin-bound state during early embryo development.

### Long, not short binding of cohesin and CTCF is associated with chromatin binding

While analysis of continuous movies reveals the overall bound fraction of molecules, we used an interlaced time-lapse microscopy (ITM) approach to further distinguish chromatin-bound molecules within different binding time classes[52]. We chose a repetitive ITM illumination pattern comprising 11.7 ms images interlaced with alternating dark times of 0.2 s and 4 s, which enabled us to separate short-bound molecules that survived a 0.2 s dark time from long-bound molecules that survived at least two 4 s dark times (>8 s) while being confined to a small area (Fig. 1f, Supplementary Movie 3 and 4). Importantly, due to the Poissonian nature of molecular interactions, this temporal scheme allowed separating binding classes with distinct dissociation kinetics[52].

Our ITM approach revealed that the fraction of short-bound molecules of both HT-Rad21 and HT-CTCF increased ca. 1.5-fold from (7.4 ± 1.8)% (cohesin)/(6.2 ± 2.5)% (CTCF) at 64-cell stage to (10.3 ± 3.8)% (cohesin)/(11.4 ± 2.5)% (CTCF) in shield stage embryos (Fig. 1g, h, right; Supplementary Fig. 9, right). The fraction of long-bound HT-Rad21 molecules increased ca. 21.5-fold from (0.6 ± 0.4)% to (12.9 ± 5.5)% during this time period, while the fraction of long-bound HT-CTCF molecules increased only ca. 3.2-fold from (2.0 ± 1.3)% to (6.2 ± 2.6)% (Fig. 1g, h, left; Supplementary Fig. 9, left).

To shed further light onto the nature of the short and long binding classes of HT-Rad21 and HT-CTCF, we performed several control measurements. First, we repeated the ITM experiments with HaloTag alone. For this HT control, short-binding events were up to 4-fold reduced and long-binding events were reduced on average by 98 % of HT-Rad21 or HT-CTCF (Supplementary Figs. 10 and 11). Second, we imaged an HT-tagged triple mutant of Rad21, HT-Rad21-3x, which cannot form a DNA-binding cohesin complex anymore[46,67,68]. We also imaged an HT-tagged CTCF mutant lacking the intermediate zinc finger domains 4 to 7, HT-CTCF-ΔZF4-7, which impedes binding of CTCF to its core motif[69–71]. We confirmed impaired chromatin binding of HT-Rad21-3x and HT-CTCF-ΔZF4-7 by subcellular protein fractionation (Supplementary Fig. 12). Both HT-Rad21-3x and HT-CTCF-ΔZF4-7 still exhibited a fraction of molecules in the short binding class (short-bound fraction) comparable to the HT-tagged wild-type proteins, while their fraction in the long binding class (long-bound fraction) was, on average, reduced by 93% (Fig. 1g, h; Supplementary Fig. 13).

Since Rad21 interacts with both its N- and C-terminus with the SMC1/3 dimer, we further tested whether the position of the HaloTag affected the bound fraction of Rad21. We thus repeated our ITM measurements with a C-terminally Halo-tagged Rad21 construct (Rad21-HT) until the shield stage (Supplementary Figs. 14 and 15). Rad21-HT bound was comparable to the N-terminally tagged HT-Rad21 at most developmental stages and showed a similar trend for both the long- and the short-bound fractions. For CTCF, N-terminal tagging has been previously established and was reported to be preferable[46,72,73].

Overall, our ITM results indicate that the long-bound fractions of HT-Rad21 or HT-CTCF are predominantly associated with chromatin binding, while short-bound fractions are presumably associated with

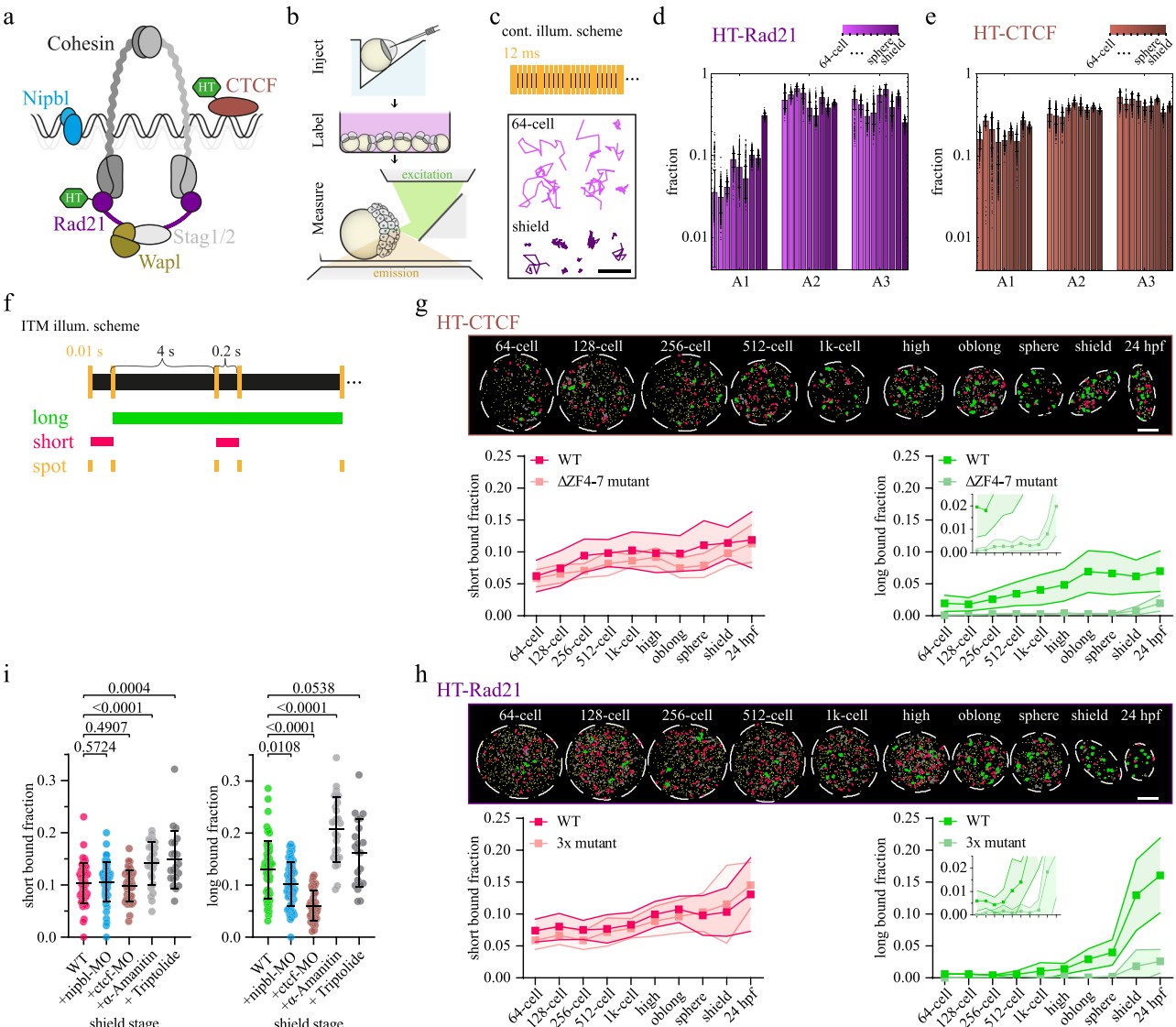

**Fig. 1 | Cohesin and CTCF bind chromatin increasingly more efficient during zebrafish development. a** Scheme of the cohesin complex. **b** Scheme outlining the workflow of zebrafish measurements. **c** Top: Scheme of continuous illumination. Bottom: Example single-molecule tracks of the 64-cell stage (bright pink) and shield stage (dark pink). Scale bar: 1 μm. Fractions of a three-component diffusion model fitted to jump distance distributions of (**d**) HT-Rad21 and (**e**) HT-CTCF. Colors indicate stages of development (64-, 128-, 256-, 512-, 1k-cell, high, oblong, sphere, shield). Bars represent mean values ± s.d. from 500 resamplings using 80% of randomly selected jump distances. For fits and diffusion coefficients see Supplementary Fig. 6-8 and statistics in Supplementary Table 1. **f** Scheme of interlaced time-lapse microscopy (ITM). Molecules are classified into different binding classes. Long (green): tracks surviving at least two long dark times (8.2 s). Short (red): Tracks surviving one short dark time (0.2 s). Spot (yellow): Molecules only detected in a single frame. Top: Example nuclei of (**g**) HT-Rad21 and (**h**) HT-CTCF ITM movies

with tracks colored according to the binding classes in panel (**f**). Grey tracks survived only one long dark time. Scale bar: 5 μm. Bottom: Fractions of short and long binding events of (**g**) HT-Rad21 (WT) or HT-Rad21-3x mutant and (**h**) HT-CTCF (WT) or HT-CTCF-ΔZF4-7 mutant. Data represent mean ± s.d. of movie-wise determined fractions. Insets show zooms into the respective graphs. Lines serve as guides to the eye. Statistics and p-Values are provided in Supplementary Table 2–7. Raw data plots are provided in Supplementary Fig. 9. **i** Fractions of short (left) and long (right) binding events of HT-Rad21 (WT) in the presence of morpholinos (MO) or RNA-polymerase inhibitors in the shield stage. Data represent mean ± s.d. of movie-wise determined fractions. All developmental stages are shown in Supplementary Fig. 14 and 16. P-values were calculated using two-sided Multiple Mann-Whitney test with a false discovery rate set to 1% (Supplementary Tables 8, 9). Source data are provided as a Source Data file for Fig. 1d, e, **g**–**i**.

unspecific transient modes of binding to DNA, RNA, proteins, or other biomolecules, or transient spatial confinement. Notably, the percentage of long-bound molecules is significantly increased in the shield stage compared to the 64-cell stage, demonstrating that chromatin binding of architectural proteins is dynamically modulated during development.

## Nipbl, CTCF, and RNA Polymerase II influence the long-bound fraction of cohesin

Cohesin relies on various cofactors, including Nipbl, which is involved in the establishment of cohesion[74–77] and supports its ATPase

activity[17–20], the cohesin unloader Wapl[21,78,79], and CTCF, which protects cohesin unloading by Wapl[33,34,80]. We hypothesized that reducing the expression level of Nipbl or CTCF should decrease the long-bound fraction of Rad21 but not affect the short-bound, unspecific fraction. We, therefore, repeated our ITM measurements in the presence of established morpholinos (MOs) impeding the expression of either Nipbl or CTCF (Fig. 1i)[81–83]. We focused on the shield stage since treatment with MOs is expected to be less effective at earlier stages due to the maternal supply of proteins. Moreover, the comparison of large bound fractions is more robust than for the small bound fractions

observed at earlier stages. We confirmed normal development of nipbl-MO injected embryos until shield-stage and severe developmental defects at 48 hpf (Supplementary Fig. 16), as reported previously[81]. Upon injection of ctcf-MO, the expression level of endogenous CTCF protein was reduced to $(15.6 \pm 2.0)\%$ at the time point of shield stage in untreated embryos (Supplementary Fig. 17). Moreover, we confirmed a developmental delay in shield-stage and lethality at later stages in ctcf-MO injected embryos, as previously reported (Supplementary Fig. 18)[53,84].

Both injections of nipbl-MO or ctcf-MO significantly reduced the long-bound fraction of HT-Rad21 at the time point of shield stage in untreated embryos, compared to untreated embryos or control-MO injected embryos (Fig. 1i, for earlier stages and control-MO measurements, see Supplementary Fig. 19 and 20). Short-bound fractions were not changed (Fig. 1i). This is expected for nipbl-MO due to disturbed loading and extrusion at reduced Nipbl levels. The presence of ctcf-MO might reduce the bound fraction due to decreased protection of cohesin from Wapl, but we cannot exclude a contribution due to the developmental delay compared to untreated embryos. Together, our measured influence of the cofactors Nipbl and CTCF on HT-Rad21 binding was compatible with their reported effects on cohesin.

Previous studies suggested that the initial formation of chromatin architecture in *Drosophila* and mice occurs independent of transcription[37,39]. On the other hand, an influence of RNA Polymerase II on the function of cohesin has been reported[85–88]. We therefore tested the impact of RNA Polymerase II, in particular RNA polymerase II elongation, on the binding of cohesin. Therefore, we repeated our ITM measurements in the presence of α-Amanitin or Triptolide (see Methods), which both prevent RNA Polymerase II elongation[39,89–93]. As expected, the development of embryos was arrested after ZGA, similar to previous observations[94–100] (Supplementary Fig. 21). Both drugs increased the short- as well as the long-bound fractions of HT-Rad21 at the time point of shield stage in untreated embryos (Fig. 1i, for earlier stages, see Supplementary Fig. 22 and 23). Increased long binding of HT-Rad21 indicates that elongation of RNA Polymerase II might disturb the binding of cohesin molecules to chromatin.

## Cohesin bound long to chromatin shows reduced spatial confinement in the presence of Nipbl

It has been shown by monitoring histone mobility and by simulations that the overall mobility of chromatin increases if the expression level of cohesin is reduced[101–104]. In our experiments, the DNA binding-deficient mutant HT-Rad21-3x showed predominantly short binding events (Fig. 1h), suggesting that long-bound HT-Rad21 mirrors the motion of chromatin. Based on these previous observations, we hypothesized that the long-bound fraction of cohesin contributed to restricting chromatin movement.

To test our hypothesis, we measured the spatial confinement of short- and long-bound HT-Rad21 in the shield stage. We devised an illumination scheme, time-lapse alternated with continuous intervals (TACO), which combined our ITM scheme to filter for short- or long-bound molecules with periods of ten continuously illuminated images. We analyzed the confinement of HT-Rad21 molecules in short or long-binding classes by quantifying the mean jump distances of tracked molecules (Fig. 2a, Supplementary Movies 5–8 and Methods). Supporting our hypotheses, we observed that HT-Rad21 bound long to chromatin exhibited significantly shorter mean jump distances of $(91 \pm 47)$ nm compared to short-bound HT-Rad21 with $(166 \pm 73)$ nm (Fig. 2b).

To further shed light on the mechanism with which long-bound cohesin restricted chromatin mobility, we reduced Nipbl levels by injecting nipbl-MO. Previous publications demonstrated a reduced association of cohesin to chromatin upon lower Nipbl levels[105–109]. In addition, lower Nipbl levels reduced the ATPase activity of cohesin[17–20]. As expected, mean jump distances of long-bound HT-Rad21 increased

significantly to $(112 \pm 58)$ nm upon nipbl-MO injection (Fig. 2b), while the short-bound fraction showed no significant difference $(174 \pm 8)$ nm compared to undisturbed conditions. Increased chromatin mobility was not due to a developmental delay of nipbl-MO-injected embryos (Supplementary Fig. 16). Our observation is compatible with the notion that reduced levels of chromatin-bound cohesin contribute to increased chromatin mobility. This might be due to less cohesin-mediated cohesion or less Nipbl-facilitated ATPase activity and, thereby, potentially loop extrusion of long-bound cohesin. (Fig. 2c).

We next repeated the TACO experiment for two additional phases during early development, where the first phase comprised the 64- to 512-cell stages (pre-ZGA) and the second phase the high to sphere stages (post-ZGA). Mean jump distances of both short-bound and long-bound cohesin molecules decreased from pre-ZGA to the shield stage (Fig. 2b). Notably, within every developmental phase, chromatin loci bound long by HT-Rad21 were significantly less mobile than short-bound HT-Rad21 molecules. The decrease in mobility of HT-Rad21-bound chromatin is compatible with loops and TADs becoming more prominent during embryo development[40].

## Long-bound cohesin shows distinct nuclear distribution

We further characterized the nuclear distribution of long-bound cohesin. Therefore, we devised a measure of radial nuclear position, the center-border distance (CBD), which ranged from the nuclear center (CBD = 0) to the periphery (CBD = 1) in five equidistant radial steps (see Methods). We then sorted every long-bound track from both ITM (Fig. 1f) and TACO (Fig. 2a) measurements into CBD bins (Fig. 2d and Supplementary Fig. 24) and normalized to bin area (Fig. 2e). We found that the density of long-bound cohesin was highest at the nuclear center in the pre-ZGA phase but shifted towards a more isotropic distribution in the post-ZGA phase. At the shield stage, we observed that long-bound cohesin was sparse at the nuclear periphery.

## Changes in the binding kinetics of cohesin and CTCF facilitate chromatin binding during development

Our ITM experiments enabled comparing chromatin binding of HT-Rad21 and HT-CTCF at multiple experimental conditions. However, they yield relative, not absolute binding times and fractions of binding classes[52]. To obtain absolute bound fractions and dissociation rates (Fig. 3a), we employed a classical time-lapse illumination approach in which we imaged HT-Rad21 or HT-CTCF molecules either continuously with 502 ms frame time or interspersed with a dark time of 4 s (Fig. 3b). This approach allows correcting for tracking errors and photobleaching of fluorophores and enables covering a broad temporal bandwidth[57,58]. Here, we counted fluorescent molecules as bound if they were confined for at least three frames (Fig. 3c and Methods) and collected the fluorescence survival times of such binding events (Fig. 3d, e). Again, we considered the three developmental phases, pre-ZGA, post-ZGA, and shield stage, which revealed a shift of fluorescence survival time distributions to longer times at later stages. We analyzed the fluorescence survival time distributions using the GRID method, which applies a Laplace transformation to infer full dissociation rate spectra of the respective survival time distribution (Fig. 3f, g; event spectra in Supplementary Fig. 25)[110,111]. Dissociation rates can be converted to residence times by inversion (see Methods).

For both cohesin and CTCF, dissociation rate spectra showed several clusters. The residence times corresponding to the smallest rate cluster increased in value and fraction during early development (Fig. 3f, g), up to comparably long average residence times of ~100 s in shield stage (cohesin: $99 \pm 6$ s, CTCF: $89 \pm 11$ s, Supplementary Table 12 and 13), with individual binding events up to ~4 min (Fig. 3d, e). In agreement with an increasing residence time, the survival times of long-bound fluorescent molecules in ITM measurements also increased during development (Supplementary Fig. 26). Informed by

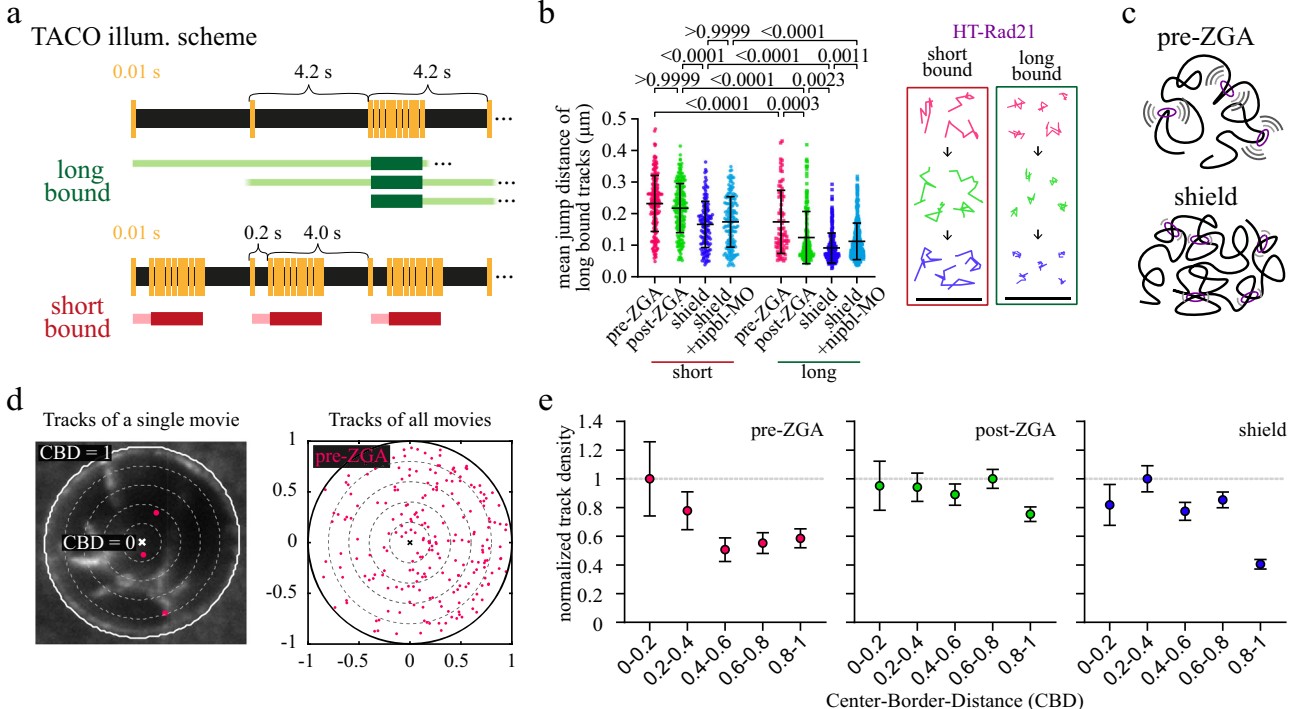

**Fig. 2 | Long-bound cohesin confines chromatin motion and shows distinct nuclear localization. a** Schemes of time-lapse illumination alternated with continuous intervals (TACO). Molecules are classified in short and long-bound binding classes comparable to ITM (Fig. 1f) and analyzed during continuous intervals (dark green and dark red bars, see Methods). **b** Mean jump distances of HT-Rad21 molecules classified as short bound (circles) and long bound (squares) at different developmental phases. pre-ZGA (red): 64-, 128-, 256, 512-cell stages pooled; post-ZGA (green): high, oblong, sphere stages pooled; shield stage (blue) and shield stage with the addition of nibpl-morpholino (MO, cyan). Insets: example tracks. Data represent mean ± s.d. *P*-values were calculated using a two-sided Kruskal-Wallis test followed by Dunn's multiple comparison test and Multiple Mann-

Whitney test with a false discovery rate set to 1% (see *p*-values in Supplementary Table 10 and statistics in Supplementary Table 11). Scale bar: 1 μm. **c** Sketch of cohesin-mediated decrease in chromatin motion. **d** Left: Example nucleus (bold white circle) subdivided into five bins (dotted lines) with the initial positions of long-bound HT-Rad21 tracks (pink). Right: Pooled initial positions of long-bound tracks of all ITM- and TACO measurements at pre-ZGA stages shown in a unit circle. For post-ZGA and shield stage, see Supplementary Fig. 24. **e** HT-Rad21 track counts per radial bin (compare panel (**d**), right) normalized to bin area and the highest bin value. Data represented as value ± statistical error (square root of track counts per bin). Statistics are provided in Supplementary Tables 6 and 11. CBD Center-Border-Distance. Source data are provided as a Source Data file for Fig. 2b, e.

our ITM measurements (Fig. 1f–h), we separated the distribution of residence times into short-bound (<10 s) and long-bound (>10 s) regimes (Fig. 3f, g, grey inserts). As expected from the ITM measurements, the overall long-bound fraction from time-lapse measurements increased moderately for HT-CTCF between pre-ZGA and shield stage, while it showed a strong increase for HT-Rad21 (Fig. 3h and Methods).

CTCF has been proposed to transiently associate with RNA-containing clusters while searching for a specific target site[49]. This process is formally similar to a search mechanism, including free diffusion and one-dimensional sliding on unspecific sites commonly applied to DNA-binding proteins[44,112]. Therefore, we considered such a mechanism to quantify the time one of the HT-CTCF molecules needed to find any of the accessible specific target sequences on chromatin. This target site search time depends on the overall bound fraction, the fraction and residence time of specific binding events, and the residence time of unspecific binding events[113,114] (see Methods). Since the DNA binding mutant HT-CTCF-ΔZF4-7 showed predominantly short binding events (Fig. 1g), we identified the long-bound fraction of HT-CTCF with specific binding. The association times we calculated for one HT-CTCF molecule to any accessible specific site varied between ~150 s (pre-ZGA), ~330 s (post-ZGA), and ~210 s (shield stage) (Fig. 3i). For cohesin, the search mechanism is not known. We, therefore, estimated the association rate from an equilibrium model including two types of binding classes of short and long cohesin binding[114] (see Methods). Similar to CTCF, the time we obtained for cohesin to find a long-bound site increased from pre-ZGA (~420 s) to post-ZGA (~902 s) and dropped to ~180 s in the shield

stage (Fig. 3i). These association times can be converted to association rates by inversion.

We further estimated the search times of any HT-Rad21 or HT-CTCF molecule to chromatin. Therefore, we included an order-of-magnitude estimate of changes in relative protein concentrations during early zebrafish development determined from published data[84] and our nuclear volume measurements (Supplementary Fig. 27 and Methods). For both HT-CTCF and HT-Rad21, the relative search time dropped during development (Fig. 3j).

The chromatin-bound fraction of proteins is coupled to the association and dissociation kinetics as well as the concentration of accessible binding sites and the nuclear size via the physicochemical law of mass action[52]. We compared the measured fraction of long-bound HT-CTCF and HT-Rad21 molecules with the bound fractions calculated from the association and dissociation rates within the physicochemical model and found good agreement (Fig. 3h and Methods). This indicates that the altered binding kinetics are sufficient to explain the increase in the bound fractions of HT-Rad21 and HT-CTCF during zebrafish development (Fig. 3k).

**Polymer modeling recapitulates the emergence of loop extrusion patterns caused by changing cohesin kinetics**

Our measurements revealed a notable increase in the residence time and a decrease in the effective search time of both cohesin and CTCF during zebrafish embryogenesis (Fig. 3). To obtain insight into whether the experimentally observed changes in binding kinetics of cohesin and CTCF could support a trend toward more structured

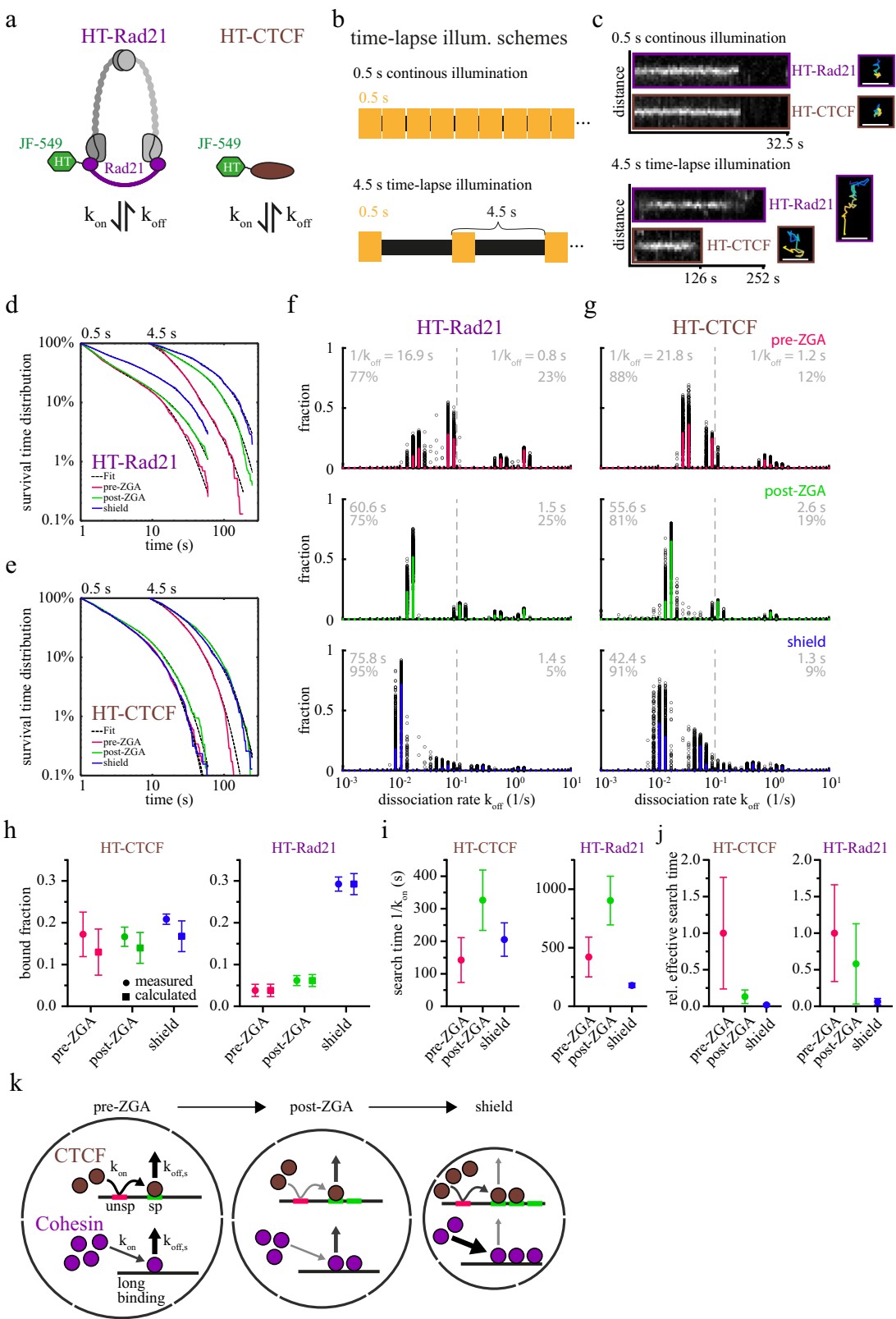

chromosomes as observed in Hi-C experiments[40], we turned to polymer simulations of cohesin-mediated loop extrusion. We used a physical model enabling the comparison of simulated chromosome interaction features against published Hi-C experimental data[40]. Our model was constructed as a polymer chain, including volume interactions and loop extrusion[103]. Within this model, we established a direct correspondence between spatial and temporal units (see

Methods). Spatial units are important for the spatial organization of chromosomes, while time units are crucial to evaluate the processivity rate of extrusion. We calibrated temporal units based on measurements in mouse embryonic stem cells[103].

To assess the effect of changing cohesin kinetics, we chose an extrusion speed of 0.5 kb/sec (an intermediate value between in vitro measurements[17,18] and in vivo estimates[73]) and performed an extensive

**Fig. 3 | Binding kinetics of cohesin and CTCF change during zebrafish embryogenesis. a** Scheme depicting the rate constants of chromatin association ($k_{on}$) and dissociation ($k_{off}$) of HT-Rad21 and HT-CTCF. **b** Schemes of continuous illumination. **c** Left: Kymographs propagating with the spot-center of continuous and 4.5 s time-lapse illumination. Right: Corresponding tracks, color-coded according to time. Scale bar: 1 μm. Survival time distributions of (**d**) HT-Rad21 and (**e**) HT-CTCF of different time-lapse conditions indicated on the top and the respective GRID fits (dashed lines). pre-ZGA (red): 64-, 128-, 256, 512-cell stages pooled; post-ZGA (green): high, oblong, sphere stages pooled; shield (blue). Statistics are provided in Supplementary Table 14. GRID state spectra of dissociation rates of (**f**) HT-Rad21 and (**g**) HT-CTCF using all data (solid line, colored according to stages) and 500 resampling runs with randomly selected 80% of data (black spots) as an error estimation of the spectra. Grey insets: Average residence times and percentages of dissociation rates larger or smaller than 0.1 s$^{-1}$ (dashed line). Detailed residence times and fractions are provided in Supplementary Tables 12–13. **h** Measured bound fractions (circle) and bound fractions calculated from kinetic rates (square) of HT-CTCF (left) and HT-Rad21 (right). Data represented as value ± s.d. (Gaussian error propagation (see Methods). Statistics are provided in Supplementary Tables 1 and 14. **i** Search times (inverse on-rates) for a single HT-CTCF or HT-Rad21 molecule to find any binding site (Methods and Supplementary Tables 15–16). **j** Relative effective search times for any HT-CTCF or HT-Rad21 molecule to find any specific binding site. Data represented as value ± s.d. (Gaussian error propagation). Calculated values in (**h**)–(**j**) are based on data in (**d**), (**e**) and Fig. 1d, e. Statistics are provided in Supplementary Tables 1 and 14. **k** Scheme depicting the concentration model: During early embryo development, the nuclear volume (outer black circle) decreases, CTCF- (brown) and cohesin (purple) molecule counts, and accessible CTCF binding sites increase. Changes in kinetic rates are depicted as variations in arrow width and color. unsp: unspecific, sp: specific binding. Source data are provided as a Source Data file for Fig. 3d–j.

sweeping of the parameter space of cohesin binding kinetics, covering one order of magnitude of association rates and residence times in the range of our measured values (Supplementary Fig. 28a, b). From this, we selected a subset of relative association rates, 0.6, 3, and 15 [Mb*min]$^{-1}$, for pre-ZGA, post-ZGA, and shield stages, whose ratios were comparable to the experimentally determined changes in the effective cohesin association rate (Fig. 3j). We also calculated processivity[115], yielding values from 25 kb in pre-ZGA and 50 kb in shield stage. For cohesin, we obtained density values on chromosomes of 0.7 Mb$^{-1}$ in pre-ZGA and 33.3 Mb$^{-1}$ in the shield stage, comparable to other experimental estimates of 2.5–5 Mb$^{-1}$ [46,73]. Our simulations focused on the pre-ZGA and shield stages with the two cohesin residence times of 50 s and 100 s, respectively (Supplementary Table 12).

We represented our simulated trajectories as contact maps, which resemble the output of Hi-C experiments that capture contacts between pairs of DNA loci (Fig. 4a, b)[40,116]. To reveal the general principles of the spatial chromosome organization, we determined the average contact probability $P(s)$ as a function of the genomic distance $s$ for each stage (Fig. 4c, d). As expected from theoretical analysis of looped polymers[117], increasing the extruder association rate and residence time resulted in increased $P(s)$ around the average size of the extruded loops (Fig. 4c) and depletion of contacts at longer length scales (orange arrowhead). This is reflected in the local slope of $P(s)$ being smaller around the average loop size and bigger at long length scales (Fig. 4e and Supplementary Fig. 28c, d). These results are in qualitative agreement with experimentally observed changes in contact maps of the pre-ZGA and shield stages (Fig. 4d, f).

To estimate the influence of increasing CTCF binding, we conducted simulations with a lower spatial resolution of 10 kb. This enabled us to simulate a longer chromosome region of 50 Mb and thus compensate for the sparsity of accessible CTCF sites in the initial phases of development (Supplementary Fig. 28e, f). We included CTCF sites identified in ChIP-seq data for the pre-ZGA and shield stages (see Methods)[94,118]. Again, in qualitative agreement with published experimental Hi-C data, we observed progressively increased insulation around CTCF sites, which is indicative of increased stalling of loop-extruding cohesin (Fig. 4g).

Comparison of polymer simulations with Hi-C data thus supports the notion that increased binding efficiency of cohesin and CTCF molecules during early development is accompanied by increased loop extrusion activity and physical insulation upon stalling of cohesin at CTCF sites (Fig. 4h).

## Discussion

Here, we characterized the binding properties of the architectural proteins cohesin and CTCF during early development in live zebrafish embryos. We initially set out to understand the nature of single-molecule binding events of CTCF and cohesin (Fig. 1). For CTCF, a comparison of the wild-type protein with the DNA binding-deficient mutant CTCF-ΔZF4-7 and an HaloTag control in ITM experiments enabled associating binding events longer than 10 s with binding to specific CTCF target sequences. The Rad21-3x mutant, which prevents DNA binding of the cohesin complex, significantly reduced binding events of cohesin longer than 10 s in ITM experiments, while shorter events were unaffected. This enabled associating long cohesin interactions with DNA binding. The origin of short binding events of CTCF and Rad21 were less clear but presumably represented unspecific transient interactions to DNA, RNA, proteins, or other biomolecules, or transient spatial confinement. To identify the origin of binding events, ITM is advantageous over continuous illumination[33] since it enables separating binding time classes and thereby distinguishing long binding events of molecules above a transient background. Overall, our experiments enabled discerning CTCF bound to specific target sequences and cohesin molecules bound to chromatin.

An important function of cohesin is sister chromatid cohesion by cohesin molecules stably connecting both daughter strands after replication until early mitosis[14,15]. The question arises whether the smallest fraction of ~0.4% of long-bound cohesin molecules we observed at early developmental stages is compatible with this function. If all these long-bound cohesins participated in cohesion, and assuming at least 100 cohesive cohesins for each of the 50 diploid chromosomes for proper chromosome segregation at nuclear division[45,119], approximately 1.1 million cohesins would need to reside in the nucleus. This value is larger than a previous estimate of 508,000 nuclear Scc1 cohesins in Prometa phase-synchronized HeLa cells[45] and an estimate of approximately 109,000 Rad21 molecules in mESC cells[73]. However, in zebrafish, Rad21 is maternally provided to compensate for transcriptional quiescence in the embryo before ZGA, which starts after ca. ten cell divisions[40,54]. Maternally provided proteins typically exceed the need of a single cell, as they are diluted by successive cell divisions. Such a mechanism may account for the very low bound fraction observed in early developmental stages.

Several of our observations and experiments suggested that long Rad21 binding events were at least partially associated with the loop extrusion function of cohesin: (i) The number of cohesive cohesin molecules required per cell for sister chromatid cohesion is not expected to change during development. Thus, even if all long-bound HT-Rad21 molecules we observed in early developmental stages were involved in cohesion, the increase in the bound fraction during development suggests at least partial involvement of Rad21 in functions of cohesin other than cohesion. In particular, as cell cycles lengthen and G1 phases appear around the 1000-cell stage after ZGA[120,121], the loop extrusion function of cohesin likely becomes more prominent, consistent with our observation of a strong increase in cohesin bound fraction. (ii) Reducing the expression level of CTCF significantly reduced long but not short Rad21 DNA bound fractions in ITM experiments (Fig. 1i). This could be due to a stabilizing effect of CTCF bound to TAD borders on cohesin through their mutual binding

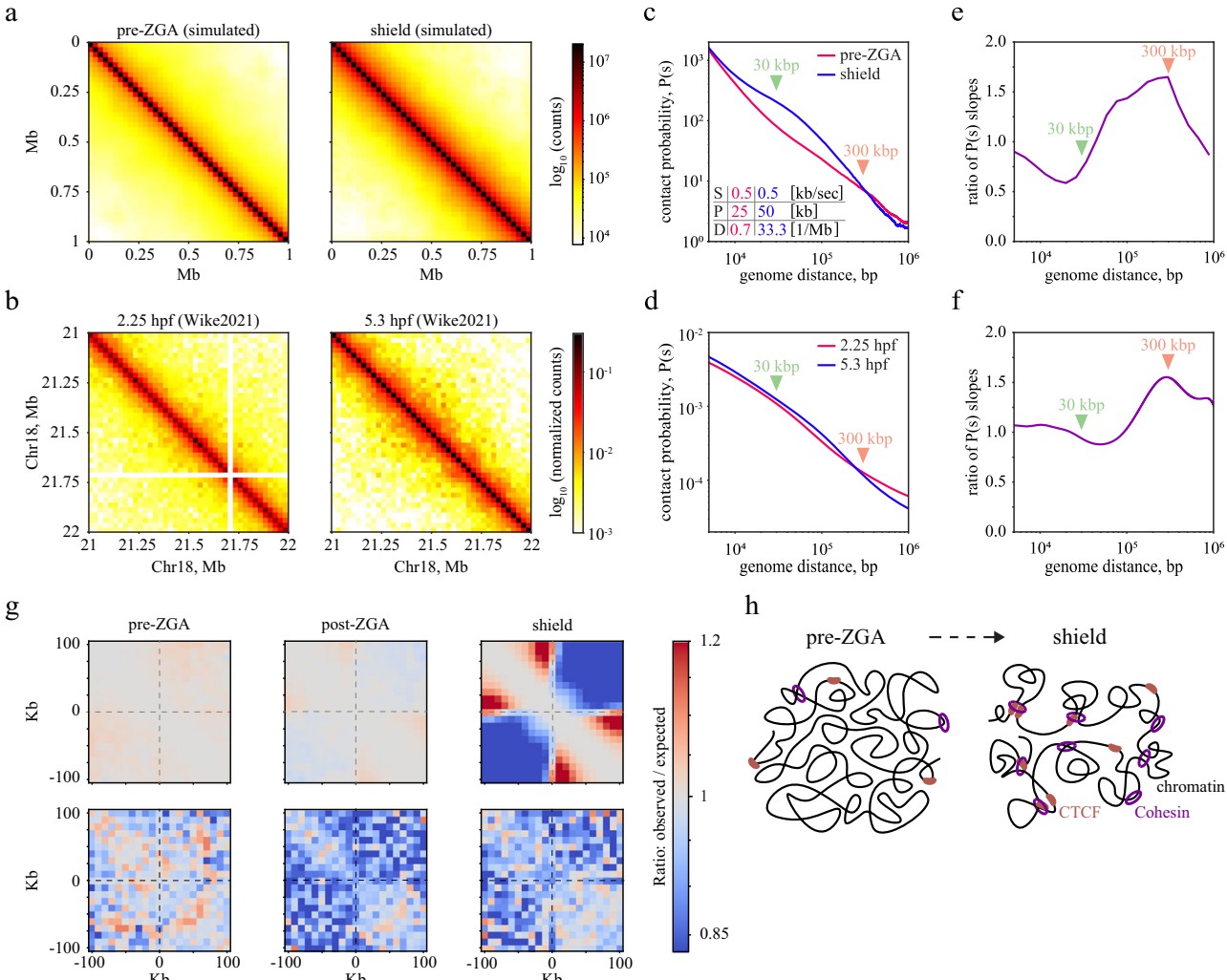

**Fig. 4 | Simulated contact maps including single-molecule data recapitulate experimental contact maps. a** Contact maps of a polymer model with 1 kb mapping for pre-ZGA (left) or shield stages (right). Data is averaged across 10 independent runs. **b** Contact maps derived from a published Hi-C dataset[40] for 2.25 hpf (hours post fertilization, similar to pre-ZGA) and 5.3 hpf (similar to shield). **c** Contact probability *P(s)* derived from the maps shown in (**a**) as a function of genomic distance. The table indicates simulation parameters for Speed of extrusion (S), Processivity of an extruder (P), and Density of extruders on chromatin (D) in pre-ZGA (red) and shield stage (blue). **d** Same as panel (**c**) but for Hi-C contacts maps from panel **b**. **e** Ratio of P(s) slopes between the shield and pre-ZGA stages using modeling data from panel (**c**). **f** Ratio of P(s) slopes between 5.3 hpf and 2.25 hpf for slopes of *P(s)* from panel **d**. **g** Pile-up plots representing average observed contact probabilities divided by expected (average contact probabilities at the same genomic distance) in +/−100kb windows around accessible CTCF sites for different developmental stages. The top row represents modeling, and the bottom row is experimental Hi-C data. **h** Scheme depicting the emergence of cohesin-mediated loops and CTCF binding due to altered binding kinetics of both species during early zebrafish development.

interaction[33]. Additionally, CTCF protects cohesin from Wapl-induced unloading[33,34,80]. However, we cannot exclude a contribution due to the developmental delay compared to untreated embryos. (iii) Mobility of Rad21 bound long to chromatin in TACO experiments was significantly increased at reduced expression levels of Nipbl (Fig. 2b). In the case of nipbl-MO treatment, we did not observe a developmental delay at shield stage. Since Nipbl reduces chromatin-bound cohesin[105–109] and stimulates the ATPase activity of cohesin[17–20], a reduced cohesive function and reduced loop extrusion activity might contribute to increased chromatin mobility. A low mobility of Rad21-bound chromatin in the presence of loop extrusion is consistent with polymer simulations[103].

We found that the binding properties of cohesin and CTCF changed during embryo development and affected chromatin architecture in several ways.

First, we observed a decrease in mobility of cohesin-bound chromatin during early zebrafish embryogenesis (Fig. 2b). This is consistent with higher chromatin mobility in pluripotent embryonic

stem cells than in differentiated cells[122,123] and compatible with an increase in chromatin structures such as compartments, TADs, or loops during embryo development[40,41]. For the radial distribution of long-bound, potentially loop-extruding cohesin, we found a transition from a high central density in pre-ZGA to a more isotropic density with a pronounced decrease at the nuclear border in shield-stage embryos (Fig. 2e). In mammals, the chromatin of differentiated cells is enriched in active histone marks at the nuclear center and repressive marks at the nuclear periphery[124,125]. Moreover, it was found that CTCF and Rad21 levels were decreased in heterochromatin-enriched lamina-associated domains of mouse embryonic stem cells[126]. Since histone modifications and chromatin accessibility are gradually established over the zygotic genome activation in zebrafish[127,128], our observations might reflect the transition from pluripotent cells to more differentiated cells.

Second, we observed an increase in the chromatin-bound fractions of cohesin upon Triptolide or α-Amanitin treatment (Fig. 1i). Since both Triptolide and α-Amanitin show distinct inhibitory effects

on RNA PolII[89–93], this increase might be due to more efficient loading at sites of stalled PolII[87,88] or due to negative interference of elongating RNA PolII with DNA-bound cohesin. The latter case is still compatible with work showing that PolII is able to push cohesin[129,130], if a fraction of cohesin remains bound to DNA, and compatible with counteraction of loop extrusion and formation of new TADs by RNA-PolII[87,131]. Our experiments suggest that disturbing transcription should not interfere with cohesin-mediated chromatin organization. This agrees with recent observations where the initial formation of chromatin architecture in *Drosophila* and mice was independent of transcription[37,39].

Third, we found that the binding kinetics of cohesin and CTCF changed during embryogenesis. For both molecular species, the effective association time decreased during development (Fig. 3k). This can be understood by an increase in the absolute protein numbers[84], a decrease in the nuclear size, and, for CTCF, an increase in the number of accessible target sites[118]. The longest residence time clusters of both species increased during early embryo development, up to similar values of ~100 s in shield stage (Fig. 3f, g), with individual binding events up to ~4 min. For CTCF, the longest average residence time is comparable to a previously reported value of ~120 s[46]. For cohesin, our value is considerably shorter than previous estimates of up to ~20 min from fluorescence recovery after photobleaching experiments in mammalian cells[46,50,80,103]. This might be due to species-specific differences. For example, longer cell cycles of ~24 h in mammalian cells compared to ~20 min in zebrafish embryos at pre-ZGA stages[121,132] might allow longer residence times. However, due to experimental limitations such as photobleaching and movement of nuclei in the living embryo, we cannot exclude that a fraction of molecules stay bound for longer times. The longest average residence times we obtained from GRID analysis should represent a lower limit. The increase in average residence times we observed during early development might be due to a mutual stabilization of CTCF and cohesin at TAD borders[33], changes in the relative concentrations of Stag1 and Stag2 that were suggested to give rise to differently stable cohesin binding[80,133], changes in the degree of acetylation of SMC3[80,134–136], the level of other interaction partners such as Wapl, or changes in the activity of enzymes such as chromatin remodelers[137–139]. It will be interesting to screen for the origin of the increasing residence times in the future.

Fourth, for both cohesin and CTCF, the fraction of long-binding events significantly increased during early embryo development. We could recapitulate this increase by inserting our observed changes in the binding kinetics of both species into a physicochemical model of binding considering a decreasing nuclear size and an increasing target site accessibility during early development (Fig. 3k)[52]. This model previously could explain a comparable increase in the bound fractions of the transcription factors TBP and Sox19b[52]. The monotonous change in binding kinetics and, thus, the increase in the bound fraction of long-bound cohesin and CTCF molecules imply that the absolute number of bound architectural proteins involved in organizing chromatin increases during embryogenesis. This is compatible with a gradual formation of chromatin architecture[40]. We additionally characterized the effect of changing binding kinetics of cohesin and CTCF on the organization of chromatin by considering our measured kinetic parameters in polymer simulations (Fig. 4). These revealed changes in chromatin architecture between the pre-ZGA and shield stages roughly comparable to Hi-C experiments[40]. We attribute the more pronounced increase in contact probability of our simulations to omitting factors that can pause or compete with loop extrusion activity and to using a low-density coil in a theta solvent, which has a steeper scaling of contact probability (~-1.5) than what was observed experimentally for mammalian chromosomes after depletion of loop extrusion factors (~-1)[12,117]. Our results indicate that the changes in the residence time and the search time of cohesin and, therefore, the increase of its chromatin-bound fraction are important to globally establish chromatin organization by DNA looping (Fig. 4j).

To conclude, our single-molecule measurements of cohesin and CTCF binding in live developing zebrafish embryos, in particular the different illumination schemes and altered cofactor levels, revealed the molecular basis of gradual chromatin architecture formation and provided a starting point to characterize the function of architectural proteins such as cohesin in the natural environment of a living embryo.

## Methods

### Zebrafish husbandry
Zebrafish of the Wild Indian Karyotype (WIK) were maintained by the Weidinger lab (Institute of Biochemistry and Molecular Biology, Ulm University, Ulm, Germany) following the guidelines of the EU directive 2010/63/EU, the German Animal Welfare Act and the State of Baden-Württemberg (Germany) in addition to being permitted by the Regierungspräsidium Tübingen. Zebrafish were solely kept for breeding, and all experiments exclusively used zebrafish embryos before 48 hours post-fertilization.

### Generation of wild-type HaloTag constructs, RNA synthesis and Western blot validation
We generated N-terminally tagged HaloTag (HT) constructs (HT-rad21, HT-ctcf) by overhang PCR with restriction enzyme sites for PacI, AscI, and complementary DNA sequences for *rad21a* (OTT-DART00000061990, ZFIN) or *ctcf* (NM_001001844.1, NCBI). The DNA template was cDNA isolated from 1-cell stage zebrafish embryos[52]. Primers are listed in Supplementary Table 25.

We cloned digested PCR fragments in a pCS2-2xHA-HALO-3xGS backbone comprising two HA-tags, the HaloTag, a flexible 3xGS-linker upstream, and an SV40 PolyA tail downstream of the insertion site. The cloning products were amplified in DH5-α cells and isolated using the QIAprep Spin Miniprep Kit (Qiagen) according to the manufacturer's protocol. All cloning products were validated by Sanger sequencing before proceeding with RNA synthesis.

We obtained the control constructs (HT-rad21-3x, HT-ctcf-ΔZF4-7) by site-directed mutagenesis of the HT-rad21 and HT-ctcf plasmids using the Q5 Site-Directed Mutagenesis Kit (New England BioLabs) according to the manufacturer's protocol with primers given in Supplementary Table 25. The HT-rad21-3x construct contained three point mutations (F609R, L613R, Q625K) that were previously described to prevent proper cohesin complex formation[46,67,68]. The HT-ctcf-ΔZF4-7 construct was depleted of its zinc finger domains 4-7 (annotation from Q6JAA4, Uniprot), responsible for CTCF's primary DNA binding to the core CTCF motif[69,70,140]. Successful mutagenesis was validated by Sanger sequencing before proceeding with RNA synthesis.

We synthesized RNA using the mMESSAGE mMACHINE SP6 Transcription Kit (Invitrogen) following the manufacturer's protocol after plasmid linearization with NotI-HF (New England BioLabs). RNA cleanup was performed with the RNeasy Plus Mini Kit (Qiagen) according to the manufacturer's protocol. α-Bungarotoxin RNA was synthesized as described above using the pmtb-t7-alpha-bungarotoxin plasmid, a gift from Sean Megason (#69542, Addgene)[141]. eGFP-lap2β mRNA was generated as described previously[52]. Control HaloTag mRNA was produced from the pCS2-2xHA-HALO-3xGS plasmid without any insert.

Ectopic protein expression levels were determined by Western blotting of samples obtained and injected on three independent days. We isolated protein from 20 shield-stage wild-type embryos and embryos injected with 6.7 pg of mRNA encoding for HT-rad21 or 10 pg of mRNA encoding for HT-ctcf based on a standard Western Blot protocol[142]. We chose 10-fold larger injection amounts compared to our single-molecule measurements to observe the HT-protein band. The primary anti-Rad21 antibody (ab992, Abcam) was diluted 1:500, and the anti-CTCF antibody (ab128873, Abcam) 1:3000. Quantification

was performed with Image Lab 6.0 by dividing the background subtracted HT-protein band intensity with the endogenous protein band intensity. Since we injected 10-fold less mRNA for single-molecule measurements, we divided the value by a factor of 10 to obtain the expression levels in single-molecule experiments. WT lysate was used as a control and to identify the HT-protein band.

## Sample preparation for single-molecule imaging

After spawning, we dechorionated zebrafish embryos in a 1 mg/ml pronase (10165921001, Sigma-Aldrich) Danieau's medium solution for three minutes, followed by washing in Danieau's medium. Still in the 1-cell stage, embryos were injected with 5 pg of eGFP-lap2β mRNA together with either 1 pg HT-ctcf, 1 pg HT-ctcf-ΔZF4-7, 0.67 pg HT-rad21, 0.67 pg HT-rad21-3x mRNA (33.5 pg for 24 hpf measurements), or 1 pg HT mRNA (control). Measurements at 24 hpf required an additional 50 pg of α-Bungarotoxin mRNA to prevent embryos from twitching[141]. To set the injection volume of 1 nl on the pneumatic PicoPump (World Precision Instruments), we adjusted the injection duration to yield droplets of 124 μm, determined by a stereo microscope (Olympus SZX2-ZB10) and a camera (CAM-SC50).

HaloTag proteins were labeled by transferring zebrafish embryos to glass tubes followed by incubation in 10 nM HaloTag-JF549 dye solution (50 nM for 24 hpf measurements). After 15 and 30 min past labeling, embryos were washed twice and further bred in Danieau's medium at room temperature (22 °C) or at 28 °C for shield-stage and 24 hpf measurements. We visually monitored the development of room-temperature embryos until they synchronously reached the 32-cell stage. Then, 4-6 embryos were transferred to a 170 μm thick glass bottom imaging dish (Delta T, Bioptechs) on the reflected light-sheet microscope. Embryos were measured at room temperature in Danieau's buffer while progressing in development. During measurement, we identified cell divisions and developmental stages using the eGFP-labeled nuclear membrane marker Lap2β. Movies were recorded after the fusion of karyomeres and at the largest nuclear cross-section. Measurements were concluded after sphere-stage. The shield-stage and 24 hpf embryos were taken out from the incubator 30 min before measurement, placed on the imaging dish as aforementioned, and measured for a maximum duration of 1 h.

To inhibit RNA Polymerase II, we used α-Amanitin (A2263, Sigma-Aldrich), which was dissolved in water and included in the RNA injection mix at a final 0.4 mg/ml concentration. As described above, the injection mix was injected into 1-cell stage zebrafish embryos with HT-rad21 and eGFP-lap2β mRNA. The concentration was similar to previous reports[96,143–146]. Moreover, we used Triptolide (645900, Sigma-Aldrich) dissolved in DMSO and diluted to a 5 μM solution in Danieau's medium on the day of usage. After dye incubation, around the 4- to 8-cell stage, embryos were washed and kept in a Triptolide solution for further development and imaging. Triptolide concentration and handling are comparable to previous reports[147,148]. The duration between treatment and shield stage measurement were approximately 9 hours for α-Amanitin and 8 hours for Triptolide.

## Morpholino (MO) handling and Western blot validation

All MOs targeting RNA translation were obtained from Gene Tools LLC (Philomath, USA) and coinjected with 0.67 pg HT-rad21 and 5 pg eGFP-lap2β mRNA as described above. We used 4 ng of the single ctcf-morpholino (ctcf-MO) or 1 ng of each nipbl-morpholino (nipbla and nipblb, in short nipbl-MO) per injection. Both the ctcf[83] and the nipbl-MOs[81] were previously validated with comparable injection amounts. Negative control measurements were performed by coinjection of 4 ng standard control morpholino mRNA obtained from Gene Tools LLC (Philomath, USA).

We validated the previously published ctcf-MO by Western blotting[83] of samples obtained and injected on three independent days. Therefore, we isolated protein from 20 shield-stage wild-type embryos or embryos injected with 4 ng ctcf-MO, based on a standard Western Blot protocol[142]. The primary anti-CTCF antibody (ab128873, Abcam) was diluted 1:3000, and the anti-γ-Tubulin antibody was diluted 1:100,000 (ab11316, Abcam). We quantified Western blots of three biological replicates with Image Lab 6.0 by first normalizing the background subtracted CTCF-band to the γ-Tubulin band, followed by dividing the morpholino-injected value with the wild-type value.

## Subcellular Protein Fractionation

We injected zebrafish embryos at the 1-cell stage with either 6.7 pg HT-rad21 mRNA, 6.7 pg HT-rad21 3x-mutant mRNA, 10 pg HT-ctcf mRNA or 10 pg HT-ctcf-ΔZF4-7 mutant mRNA. To ensure detection of HT-protein bands, injection amounts were increased 10-fold compared to our single-molecule measurements, consistent with our Western blot methodology. Similarly, we removed the comparably large proportion of interfering yolk proteins from 150-190 shield-stage embryos based on standard protocols[142]. The resulting cell pellet was washed once in PBS, followed by centrifugation at 500 g for 2 min. To enhance subsequent lysis, nearly all supernatant was removed followed by a short vortex on the lowest setting to gently disperse cells in the residual supernatant. Subcellular fractionation was performed using the Subcellular Protein Fractioning Kit for Cultured Cells (78840, Thermo Fisher Scientific), according to the manufacturers protocol for a 20 μL cell volume. For Western blot analysis, we used 20 μl of each extract. The primary anti-Rad21 antibody (ab992, Abcam) was diluted 1:500, the anti-CTCF antibody (ab128873, Abcam) 1:2000 and the anti-Histone H2B antibody (MA5-14835, Thermo Fisher Scientific) 1:1000. We calculated the ratio of HT-protein to endogenous protein from background subtracted bands using Image Lab 6.0.

Lane profiles for HT-CTCF-ΔZF4-7 were generated in FIJI[149]. Western blots images were first rotated to vertically align the 3-SNE and 4-SNE bands. Using a rectangular selection over a straight part of each band, we obtained intensity profiles by using the "Plot Profile" function. For each lane, data was normalized by first subtracting the minimum value of the dataset, followed by dividing by the range of the dataset (maximum minus the minimum value).

## Co-Immunoprecipitation (Co-IP) of Smc3 through HA-tagged HT-Rad21

To investigate the incorporation of our HA-tagged HT-Rad21 subunit in the cohesin complex, we performed Co-Immunoprecipitation (Co-IP) using anti-HA-Tag beads. We injected zebrafish embryos at the 1-cell stage with either 6.7 pg HT-rad21 mRNA (10-fold mRNA amounts compared to our single-molecule measurements) or left them uninjected as a wild-type (WT) negative control, comparable to our Western blot methodology. We deyolked approximately 165 shield stage zebrafish embryos of each sample based on standard protocols[142]. The resulting cell pellet was washed in PBS and centrifuged at 300 g for 1 min. To enhance subsequent lysis, nearly all supernatant was removed followed by a short vortex on the lowest setting to gently disperse cells. Cells were then lysed in ice-cold Pierce™ IP Lysis Buffer (87787, Thermo Fisher Scientific) for 5 min on ice after which 10% was taken as the Input reference. We centrifuged the remainder at 13,000 g for 10 min at 4 °C to pellet cell debris, and used the liquid phase for the following Co-IP procedure. We performed the Co-IP using the Pierce™ HA-Tag IP/Co-IP Kit (26180, Thermo Fisher Scientific) according to the manufactures protocol, incubating the anti-HA agarose beads and sample overnight, followed by Elution Protocol 2.

To detect proteins, we equally distributed the Input and Co-IP samples from injected and WT embryos across two gels and resolved the proteins simultaneously by SDS-PAGE, resulting in two equally loaded blots. To unambiguously identify the presence of each protein, we probed one blot with 1:1000 anti-Smc3 antibody (JM10-75, Thermo Fisher Scientific) and the other with 1:5000 anti-HA Tag antibody

(2-2.2.14 #26183, Thermo Fisher Scientific). To confirm the presence of both Smc3 and HA-tagged HT-Rad21 protein on a single blot, the first blot probed with anti-Smc3 antibody was subsequently probed with 1:2500 anti-HA Tag antibody.

### Reflected light-sheet microscopy

Single-molecule movies of live developing zebrafish embryos were recorded on a custom-built reflected light-sheet microscope, allowing lateral excitation of the ~700 μm large zebrafish embryos[52]. A commercial Nikon TI microscope body laid the foundation together with a 1.2 NA water-immersion objective (60× 1.20 NA Plan Apo VC W, NIKON), dichroic mirror (F73−866/F58−533, AHF), emission filter (F72−866/ F57−532, AHF), notch filter (F40−072/F40−513, AHF) and an EM-CCD camera (iXon Ultra DU 897U, Andor). An additional 1.5x post-magnification in the detection path resulted in an effective pixel size of 166 nm. Peripheral devices were controlled with a NIDAQ data acquisition card (National Instruments) and the NIS Elements software (Version 4.40.00 64-bit, Nikon). Laser illuminations during camera exposures were driven by a custom controller, reacting to camera triggers and following predefined patterns (Supplementary Table 26).

The excitation path used 488 nm (iBEAM- SMART-488-S-HP, 200 mW, Toptica) and 561 nm (Jive 300, 300 mW, Cobolt) lasers, combined and then controlled by an AOTF (AOTFnC-400.650-TN, AA Optoelectronics), followed by a single-mode fiber connected into a custom-built tower. The light sheet is created in the tower by a cylindrical lens that is focused in the back-focal plane of a vertically mounted water-dipping objective (40× 0.8 NA HCX Apo L W, Leica). The tower was placed above the imaging dish and reflected the excitation light sheet laterally with a custom-coated cantilever (40 nm Al coating on both sides of HYDRA2R-100N-TL-20, AppNano). The approximately 3 μm thick light sheet was set to 47 mW and 5 mW after the excitation objective for the 561 nm and 488 nm laser, respectively.

We recorded continuous, interlaced time-lapse microscopy (ITM) and time-lapse alternated with continuous intervals (TACO) movies with 10 ms of exposure time and 1.7 ms of camera readout time, resulting in a total frame cycle time of 11.7 ms. Movies for time-lapse illumination were recorded with 500 ms exposure and a total frame cycle time of 501.7 ms. All illumination schemes contained combinations of the nuclear membrane marker eGFP-Lap2β (488 nm) to track the nucleus and drift, 561 nm exposure for HaloTag-JF549 excitation, and dark times without laser excitation (see Supplementary Table 26).

### Analysis of single-molecule microscopy data

Single continuous movies were split into single files for the 488 nm and 561 nm channels using the Movie splitter tool of TrackIt[58]. Movies with considerable drift were discarded to prevent tracking and subsequent analysis errors.

We analyzed all single-molecule microscopy data with the nearest neighbor algorithm of TrackIt and tracking parameters shown in Supplementary Table 27. Nuclear tracking regions were drawn based on z-projections of the Lap2β signal if not indicated otherwise.

We used a custom MATLAB script to analyze the time intervals between consecutive early developmental stages (64-cell to the sphere stage) that were recorded with our ITM scheme (Fig. 1f). The script determined the time difference between consecutive stages for each embryo separately, by comparing the creation dates of the original movie files. If a stage was missing for a particular embryo, time differences for that stage were not calculated.

### Data analysis of continuous movies: bound fractions and diffusion coefficients

First, we recorded continuous movies with a frame cycle time of 11.7 ms and tracked 100 frames with TrackIt (Supplementary Table 27). Next, we determined diffusion rates and amplitudes by fitting cumulative distributions of the jump distances within detected tracks with a three-component Brownian diffusion model using the data analysis tool of TrackIt[58]. The bin width was set to 1 nm, resulting in a total of 747 bins. Jumps over gap frames were removed, and the number of jumps to consider per track was set to 10 in order to prevent over-representation of bound molecules[57]. Effects from molecules moving out of the focus plane (overestimation of bound fraction) or tracking of 3D movement in a 2D plane (underestimation of diffusion coefficients) were not corrected. However, these effects are present to a similar extent in all stages, and relative comparisons are not affected. Errors of diffusion coefficients and amplitudes were estimated based on the standard deviation from analyzing 500 subsets of the jump distances, including 80% of randomly selected jump distances. The amplitude of the slowest diffusion component represented the overall bound fraction $f_b$ of the tracked molecules. The unbound fraction of tracked molecules is given by $p_f = 1 - f_b$.

To evaluate whether a three-component diffusion model was superior to a two-component diffusion model, the reduced chi-squared and Akaike Information Criterion (AIC) were calculated for each model. The AIC provides a measure for the balance between the goodness of a fit and the complexity of a model, where more complex models with a higher number of parameters are punished. The differences between the models were calculated according to Eq. (1):

$$\Delta AIC_i = \left(2k_{2rate} + n\ln(RSS_{2rate})\right) - \left(2k_{3rate} + n\ln(RSS_{3rate})\right) \quad (1)$$

here, the residual sum of squares (RSS) reflects the sum of differences between data and fit from the cumulative jump distance plot, $k$ reflects the number of parameters from the two- or three-component models, and $n$ is the total number of bins. A positive ΔAIC value indicates a preferential three-component model, whereas a negative value indicates a preferential two-component model.

### Data analysis of interlaced time-lapse microscopy (ITM): relative fractions in binding time classes

We recorded ITM movies to classify molecules into three binding time classes, namely spots (single frame), short bound (0.2 s), and long bound (>8.2 s), and to calculate the relative fraction of molecules binding in each class. The illumination scheme includes short and long dark times between frames of molecule detection (Supplementary Table 26).

The tracking and analysis parameters for TrackIt are provided in Supplementary Table 27. Bound fractions of the binding classes were calculated movie-wise with TrackIts data analysis tool using the following Eqs. (2) and (3):

$$long\ bound\ fraction = \frac{N_{long}}{N_{all\ events}} = \frac{N_{long}}{N_{long} + N_{short} + N_{spot}} \quad (2)$$

here $N_{long}$ represented the number of all tracks surviving two or more long dark times (>8.2 s), and $N_{short}$ all tracks with one long or no long dark time. $N_{spot}$ depicted the number of all single detections.

$$short\ bound\ fraction = \frac{N_{short}}{N_{all\ events}} = \frac{N_{short}}{N_{long} + N_{short} + N_{spot}} \quad (3)$$

here $N_{short}$ was the sum of all tracks that lasted for one short dark time (0.2 s), and $N_{long}$ was the sum of tracks that passed at least one long dark time.

### Data analysis of time-lapse alternated with continuous intervals (TACO): mobility of molecules in certain binding time classes

The TACO scheme can classify tracks in binding time classes similar to ITM, namely short and long; however, the scheme adds fast and continuous tracking periods. Thus, TACO can provide additional diffusion analysis of similarly classified tracks examined with ITM.

The illumination scheme consisted of single detection frames followed by dark times (detection periods) (Supplementary Table 26) to track whether a molecule stayed bound while minimizing laser exposure and photobleaching. Tracking and analysis of molecules were facilitated by appending 10 continuous frames of 11.7 ms frame cycle time (analysis periods). We pooled multiple stages: pre-ZGA (64-cell, 128-cell, 256-cell, 512-cell), post-ZGA (high, oblong, sphere), and shield. After tracking of molecules (Supplementary Table 27), each track was split into the first complete analysis period if, in the case of the long TACO scheme, the track survived at least 2 dark times (=8.4 s) or in the case of the short TACO scheme, the track only survived a single short dark time (=0.2 s) without being detected in the following long dark time. To prevent overrepresentation of single detections, only the first analysis region without gap frames was used, and all other analysis regions of this track were discarded.

## Data analysis of time-lapse microscopy: residence times and bound fractions

We used time-lapse microscopy and our previously published genuine rate identification method (GRID) to determine residence times of the HT-rad21 and HT-ctcf constructs[110]. GRID is based on solving the inverse Laplace transformation of fluorescence survival time distributions gained from multiple time-lapse conditions.

The first time-lapse condition builds on 501.7 ms frame-cycle time movies with continuous excitation of the JF549 labeled HaloTag construct. The second condition consisted of a repeating pattern with a single 501.7 ms excitation frame followed by a dark time of 4013.6 ms without laser illumination (Supplementary Table 26). We chose 500 ms exposure time to blur out diffusing molecules and focus on tracking bound molecules. We pooled movies of both time-lapse conditions into pre-ZGA (64-cell, 128-cell, 256-cell, 512-cell), post-ZGA (high, oblong, sphere), and shield stage. Tracking radii for each time-lapse condition were determined to yield equal loss probabilities of 0.0025 in TrackIt. The loss probability, accounting for tracking errors and photobleaching, was set equally for time-lapse conditions and stages of each construct. The resulting tracking parameters were the same for all stages and are listed in Supplementary Table 27.

We used GRID to simultaneously analyze the fluorescence survival time distributions of both imaging conditions and obtain the event spectrum of dissociation rates ranging from $10^{-3}$ s$^{-1}$ to $10^1$ s$^{-1}$ (Supplementary Fig. 25). The amplitudes $A^e$ of the event spectrum of dissociation rates correspond to the relative fraction of molecular binding events occurring with a certain dissociation rate in a period of time. We transferred the event spectrum to the state spectrum of dissociation rates by dividing each amplitude with the corresponding dissociation rate and renormalization[110]. The amplitudes $A^s$ of the state spectrum correspond to the relative fraction of molecular binding events that belong to a certain dissociation rate in a snapshot of time. We estimated the error of both event and state spectra by resampling 500 data subsets, each including 80% of randomly selected survival times.

To obtain the residence times $\tau_s$ of long-binding events, we averaged all dissociation rates below 0.1 s$^{-1}$ ($k_{off,s}$) based on the binding time classification determined in ITM experiments and inverted the resulting value. The relative fraction $A^s_s$ of long-binding events is given by the corresponding averaged amplitudes of the state spectrum. To obtain the residence times $\tau_u$ of short-binding events, we performed these operations with dissociation rates above 0.1 s$^{-1}$ ($k_{off,u}$). Again, the relative fraction $A^s_u$ of short-binding events is given by the corresponding averaged amplitudes of the state spectrum. We calculated the overall fractions $p_{b,s} = A^s_s * f_b$ of long-binding events and $p_{b,u} = A^s_u * f_b$ of short-binding events by multiplying the relative bound fractions with the overall bound fraction $f_b$ obtained from the analysis of continuous movies (see above). We obtained the errors of the residence

times and fractions from the standard deviation of the resampled state spectra and Gaussian error propagation, if necessary.

## Calculation of search times

DNA-binding proteins follow a search mechanism of facilitated diffusion, which includes three-dimensional diffusion in the nucleoplasm and one-dimensional sliding along unspecific DNA, to search for their specific target sites[112]. To estimate the search time $\tau_{search}$, that is the average time a DNA-binding protein needs to find any of its specific target sites, we considered an equilibrium binding model with two different types of binding sites, unspecific and specific[114]. In this model, the DNA-binding protein performs the unspecific and specific binding reactions:

$$[T] + [D_u] \xrightleftharpoons[k_{off,u}]{k_{on,u}} [TD_u] \tag{4}$$

$$[T] + [D_s] \xrightleftharpoons[k_{off,s}]{k_{on,s}} [TD_s] \tag{5}$$

Were $[T]$ is the concentration of the free protein, $[D_u]$ and $[D_s]$ are the concentrations of free unspecific and specific binding sites, $[TD_u]$ and $[TD_s]$ are the concentrations of protein-bound binding sites, $k_{on,u}$ and $k_{on,s}$ are the bimolecular association rates and $k_{off,u}$ and $k_{off,s}$ are the dissociation rates (see time-lapse microscopy analysis). If the concentration of binding sites is much larger than that of the DNA-binding protein and the bound fractions are small, both reactions are approximately decoupled.

The dissociation constant of either reaction is given by

$$K_{d,i} = \frac{k_{off,i}}{k_{on,i}} = \frac{[T][D_i]}{[TD_i]} \tag{6}$$

with $i = u,s$. Thus, there is a relation between the ratio of kinetic rates and the ratio of the free fraction $p_f$ (see continuous movie analysis above) to the bound fraction $p_{b,i}$ (see time-lapse microscopy analysis above) of the DNA-binding protein:

$$\frac{k_{off,i}}{k_{on,i}[D_i]} = \frac{[T]}{[TD_i]} = \frac{p_f}{p_{b,i}} \tag{7}$$

We set $k^*_{on,i} = k_{on,i}[D_i]$ in the following.

Within this search model, the target site search time $\tau_{search}$ of the DNA-binding protein depends on the time spent finding and binding to an unspecific binding site and the number of unspecific encounters before a specific target sequence is found. It can be estimated to[113,114]

$$\tau_{search} = \frac{N_{trials}}{k^*_{on,u}} + \frac{(N_{trials} - 1)}{k_{off,u}} \tag{8}$$

where $N_{trial}$ is the number of unspecific encounters. It is given by

$$N_{trials} = \frac{1}{A^e_s} \tag{9}$$

with $A^e_s$ the amplitude of the long-binding events in the GRID event spectrum of dissociation rates (see time-lapse microscopy analysis).

For CTCF, which follows a search mechanism that is formally equivalent to the mechanism of facilitated diffusion[44,46], we also used this formalism. We obtained the search time $\tau_{search}$ from Eq. (8) by calculating $k^*_{on,u}$ and $N_{trial}$ with Eqs. (7) and (9) and inserting the measured parameters $A^e_s$, $k_{off,u}$, $p_f$, and $p_{b,u}$. All measured parameters are provided in Supplementary Table 15.

For cohesin, a detailed model of the search mechanism is missing. Since we observed distinct short and long binding events for cohesin similar to CTCF, we considered the same equilibrium binding model with two types of binding events. From this model, we obtained the search time $\tau_{search}$ of cohesin by calculating $k^*_{on,s}$ directly with Eq. (7), the measured dissociation rate of long binding events $k_{off,s}$ and the fractions $p_f$ and $p_{b,s}$, and taking the inverse $\tau_{search} = 1/k^*_{on,s}$. All parameters inserted in this equation can be found in Supplementary Table 16.

## Calculation of the relative effective search time

For our polymer simulations, we additionally estimated the effective search time $\tau_{effective}$ of any cohesin or CTCF molecule to find any long-binding site by including estimates of the protein concentrations. With the number of molecules $N_{mol}$ per nuclear volume $V_{nuc}$, the effective search time is given by:

$$\tau_{effective} = \tau_{search} V_{nuc} N_{mol}^{-1} \qquad (10)$$

We obtained the nuclear volumes for pre- and post-ZGA stages from the area of the cross-section given by the eGFP-Lap2β signal in HT-Rad21 ITM movies (Supplementary Fig. 27) and assuming a spherical nuclear shape[52].

For the more elliptical nuclei in the shield stage, we obtained the nuclear volume from the signal of GFP-H2B in z-scans with a Lattice Light-Sheet microscope (see section below).

The amount of molecules $N_{mol}$ at different developmental stages of zebrafish embryos were approximated from published quantification data of Western blots[84]. We used the Rad21 and CTCF protein levels of stages closest to our pooled stages, namely 2.5 hpf for pre-ZGA, 3.3 hpf for post-ZGA, and 5.4 hpf for shield stage. We normalized to the effective search time in pre-ZGA to obtain relative effects of the effective search times. Gaussian error propagation was used for error estimations.

## Calculation of long-bound fractions from kinetic rates

The overall bound fraction of short- and long-bound molecules, $f_b$ (see continuous movie analysis), is given by

$$f_b = \frac{[TD_s] + [TD_u]}{[TD_s] + [TD_u] + [T]} \qquad (11)$$

and the relative bound fraction of long-binding events, $A^s_s$ (see time-lapse microscopy analysis), by

$$A^s_s = \frac{[TD_s]}{[TD_s] + [TD_u]} \qquad (12)$$

Thus, the overall fraction of long-bound molecules, $p_{b,s}$, is

$$p_{b,s} = f_b * A^s_s = \frac{[TD_s]}{[TD_s] + [TD_u] + [T]} \qquad (13)$$

Using Eq. (7), the expressions for $f_b$ (Eq. 11) and $A^s_s$ (Eq. 12) can be expressed in terms of kinetic rates, and $p_{b,s}$ becomes

$$p_{b,s} = \frac{k^*_{on,s}/k_{off,s}}{k^*_{on,s}/k_{off,s} + k^*_{on,u}/k_{off,u} + 1} \qquad (14)$$

Notably, the on-rates $k^*_{on,i}$, $i = u,s$, include the concentration of binding sites, which depend on the number of accessible binding sites and the nuclear volume.

We compared $p_{b,s}$ obtained by Eq. (13) by inserting the measured bound fractions and $p_{b,s}$ obtained by Eq. (14) by inserting the kinetic rates.

## Lattice Light-Sheet microscopy (LLSM): nuclear volume determination

We obtained nuclear volumes for pre- and post-ZGA measurements by assuming a spherical nucleus shape and calculating the sphere volume using the nuclear membrane eGFP-Lap2β signal as the cross-sectional area. Shield stage nuclei, however, are primarily elliptical and prevent volume estimations based on the nuclear cross-section alone. Therefore, we switched to lattice light-sheet microscopy to determine the nuclear volume from image stacks of GFP-labeled histone H2B. We injected zebrafish embryos in the 1-cell stage with 18.5 pg of mRNA encoding for GFP-H2B and incubated embryos that passed the 2-cell stage at 28 °C in an incubator until the shield stage.

To account for the larger size of zebrafish embryos compared to cells, we devised a method to mount zebrafish embryos on the lattice light sheet microscope. This was necessary due to its orthogonal geometry of excitation and detection objectives[150]. We placed zebrafish embryos upright on the sample holder by mounting them in agarose wells without fully covering the embryo, thereby preventing possible aberrations. We glued a 5 mm glass coverslip to the bottom of the sample holder to form a trough. This trough was slightly overfilled with 3% of low-gelling temperature agarose (A9045, Sigma-Aldrich) dissolved in Danieau's medium. To insert wells into the agarose, we immediately placed the sample holder upside down on a custom 3D print (see Supplementary Fig. 27). The 3D print contained notches to center the sample holder and 700 μm tall vertical pins, which stamped small wells into the gelling agarose. We filled the agarose wells with Danieau's buffer and transferred a single embryo to a well with the animal cap pointing up. Ideally, the yolk reached the bottom of the well, and the animal cap protruded out of the well. This approach did not require adjustments to the sample holder setup, since the animal cap is mounted at a similar spatial location as cells.

We used a lattice-light sheet microscope custom-built according to a published design[150]. To image the nuclear histone marker GFP-H2B of shield stage embryos, GFP was excited with a dithered light sheet from a 488 nm laser (Genesis MX488 STM OPSL, Coherent Europe B.V.) that was formed by a square lattice and an annular mask with an outer NA of 0.55 and an inner NA of 0.52. We recorded image stacks scanning multiple nuclear volumes with a step size of 332 nm and 50 ms exposure time. Image stacks were deskewed and deconvolved using the lattice light-sheet post-processing utility LLSpy[151]. We then performed nuclear volume analysis in FIJI[149] by cropping image stacks to exclude frames at the start or end of a movie without a stable GFP signal, followed by equalizing irregular H2B signal with a median filter (20 px) and background subtraction (1 px). Leveraging the 3D ImageJ Suite[152], we segmented nuclei in every image stack by thresholding the H2B signal. Interphase nuclei were manually selected, and their volume analysis was done with FIJIs' integrated 3D objects counter.

## Center-Border-Distance (CBD) analysis

We determined the relative position of long-bound tracks between the center of a nucleus and its border.

We started by pooling ITM and TACO movies in three stages: pre-ZGA (64-cell, 128-cell, 256-cell, 512-cell), post-ZGA (high, oblong, sphere), and shield stage. Next, we manually identified the nuclear outline at the first, middle, and last frame of a movie based on the nuclear lamina signal (eGFP-Lap2β), and interpolated the shape between these outlines for every frame of the movie to account for cell movement. We proceeded with tracks of the long-bound classification (see ITM: Fig. 1f, TACO: Fig. 2a). Since nuclei showed different sizes and forms, we normalized the CBD of tracks between the nucleus center (CBD = 0) and border (CBD = 1). The CBD distances were determined for every track based on a vector spanning the center of mass of the current nuclear outline, the initial xy-position of a track, and the border of the current nuclear outline. Finally, we sorted track-CBDs in a histogram with five ring-shaped area bins. To account for the increasing

area of bins closer to the border, we divided the track counts of a bin with the bin area and further normalized the values by the largest histogram value. The errors represented the statistical counting error and were calculated as the square root of track counts per bin.

### Identification of accessible CTCF sites from ATAC-seq and ChiP-seq data

We identified CTCF sites in open chromatin regions to be used in our simulations.

For ATAC-seq analysis and peak calling, we used published ATAC-seq pairs end fastq reads from the Gene Expression Omnibus database (GEO) with the GEO accession number GSE130944[94]. For the analysis, the developmental stages 256-cell (SRR9032650, SRR9032651), oblong (SRR9032665, SRR9032666) and shield stage (SRR9032668, SRR9032669, SRR9032670, SRR9032671, SRR9032672, SRR9032673) were used. All ATAC-seq fastq files were analyzed according to the pipeline described in the original publication. Next, we aligned Fastq files to zebrafish reference genome danRer10 (see NCBI RefSeq assembly GCF_000002035.5) using the bowtie2 tool[153] with the parameters -X 2000 --no-mixed --no-discordant. Peak calling was performed after filtering out duplicates and low-quality alignments with $q < 30$. Finally, ATAC peaks were called using MACS2[154] with parameters --no-model --no-lambda, and a cutoff False Discovery Rate (FDR) of 5% was applied using the parameter --q 0.05. As a control (-c), a digestion of genomic DNA was used (SRR9032640). The common peaks of each replicate from each developmental stage were used for further downstream analysis.

To extract the ATAC signal, Chromosome 18 was digested in 1 kb windows with the command bedtools makewindows[155]. For each window, the signal from the bigwig of each developmental stage was calculated (256, oblong, and shield from GEO: GSE130944) with bigWigSummary.

For CTCF orientation analysis, CTCF peaks and CTCF motifs were obtained from a published data set with the GEO accession number GSE133437[118]. Each peak center was extended +/− 5 kb, and CTCF orientation per extended peak was extracted using homer[156] and the parameters −find known_zebra_motif and −size given. The CTCF motif used is given in Supplementary Table 28.

### Polymer simulations

For the polymer-chain model, we used Langevin molecular dynamics simulations of beads connected with springs implemented in LAMMPS[157]. To simulate volume interactions between beads, we utilized the Lennard-Jones potential in Eq. (15) with coefficients representing θ-solvent[158]

$$U_{LJ}(r) = 4\epsilon \left[ (\sigma/r)^{12} - (\sigma/r)^6 \right] \qquad (15)$$

Where $\epsilon$ is the potential well depth, $\sigma$ the distance at which the potential vanishes (cutoff radius), and $r$ the distance between beads. The Lennard-Jones potential prevents the chain from self-intersections.

Bonds between beads were simulated using the harmonic potential given in Eq. (16):

$$U_H = \frac{k}{2}(r - b)^2 \qquad (16)$$

Where $k$ is the spring constant, r the distance between beads and b the distance at which the potential is zero. We used a relatively high $k = 40$ that prevents stretches allowing self-intersections.

Loop extrusion was simulated as a process comprised of 4 steps:

1. Extruding molecules are stochastically loaded onto an adjacent pair of polymer beads every 400k simulation steps ( ~ 4 sec). We set the constant probability representing loading rate which we varied. Extruders are simulated as additional bonds between beads that can slide along the chain.

2. Every 400k simulation steps, each extruder makes a step to increase the size of the loop. This step results in an extrusion processivity of 0.5 kb/sec. Starting from the initial loop [i, i + 1], the extruder will bind [i-1, i + 2] in the next step and so on.

3. Every 400k simulation steps, each extruder can be unloaded with a constant probability. This probability determines the residence time of the extruder.

4. When two loops encounter each other during loop extrusion on the chain, for instance loop 1 [i, i + 10] and loop 2 [i + 11, i + 20], they cannot pass one another. The common base of loop 1 [i + 10] and loop 2 [i + 11] becomes stationary, whereas the loop can continue extruding at the other end of loop 1 [i] and loop 2 [i + 20].

We did not include CTCF and the loop-extrusion blocking activities of other complexes, such as PolII[88,159] and the MCM complex[160], nor any other features that can pause or compete with loop extrusion activity.

We had to map arbitrary units of space and time to real units. We chose 1 bead to represent 1 kb of chromatin. Considering the size of a nucleosome with a linker of ~15 nm and assuming that the chain of nucleosomes is folded in a random walk configuration, we estimated that 1 bead corresponds to $15^*\sqrt{5}$ nm $\approx$ 33.5 nm. To map simulation onto experimental time, we used previously acquired data for mouse embryonic stem cells, where we determined the time it takes for a chromosome segment of 8 kb to move about its own size[103]. We estimated that for the current system and resolution, 100k steps correspond to 1 sec. We fixed extrusion processivity at 0.5 kb/sec and we varied loading and unloading rates, as described above.

To test the effect of CTCF sites we remapped our system to 10 kb. Accessible and occupied CTCF sites were identified as described above for three developmental stages, namely 2.25, 4, and 5.3 hpf. Since the permeability of these sites to loop extrusion is a priori variable and unknown, we did not account for such variability and simulated a simplified model with impermeable sites. The time was mapped as for 1 kb simulations, yielding that 40k simulation steps corresponded to ~1 sec.

### Reporting summary

Further information on research design is available in the Nature Portfolio Reporting Summary linked to this article.

## Data availability

Source data are provided with this paper as a separate file named 'Source Data.xlsx'. All single-particle tracking data and simulation data are freely available at Dryad [https://doi.org/10.5061/dryad.3bk3j9ks8][161]. Data supporting the findings of this manuscript are additionally available from the corresponding author upon reasonable request. Source data are provided with this paper.

## Code availability

The single-molecule tracking software TrackIt is freely available on GitLab [https://gitlab.com/GebhardtLab/TrackIt] or Zenodo [https://zenodo.org/records/7092296][162]. For polymer modeling, LAMMPS with an implemented Loop Extrusion module is freely available on GitHub [https://github.com/polly-code/lammps_le].

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

## Acknowledgements

We thank Gilbert Weidinger (Ulm University) for generously granting us access to his zebrafish facility and for offering insightful discussion, as well as Doris Weber (Ulm University) for her expert guidance on zebrafish handling. Moreover, we are thankful to Astrid Bellan-Koch (Ulm University) and Jutta Hegler (Ulm University) for adding in cloning the fusion constructs and performing Western blots, Jens Michaelis (Ulm University) for his excellent contributions in developing the lattice light-sheet microscope, and members of the Gebhardt and Michaelis labs for supportive discussions. We further thank Aleksandra Galitsyna (Massachusetts Institute of Technology) for important discussions on Hi-C data. pmtb-t7-alpha-bungarotoxin was a gift from Sean Megason (Harward Medical School). The work was funded by the European Research Council (ERC) under the European Union's Horizon 2020 Research and Innovation Program (no. 637987 ChromArch to J.C.M.G.) and the Deutsche Forschungsgemeinschaft (DFG, German Research Foundation no. 422780363 SPP 2202 and no. 427512076 to J.C.M.G.). Support by the Collaborative Research Centers no. 316249678 (CRC 1279) and no. 450627322 (CRC 1506) to J.C.M.G. and the DFG Center for Translational Imaging MoMAN (no. 447235146) of Ulm University is acknowledged. Research of P.I.K. and L.G. was funded by the Novartis Foundation, the European Research Council (ERC grant no. 759366, 'BioMeTre'), and the Swiss National Science Foundation (grant no. 310030_192642). Work in the Papantonis lab was supported by the German Research Foundation (DFG) via the Collaborative Research Center 1565 (Project no. 469281184) and the Priority Program 2202 (Project no. 422389065).

## Author contributions

J.C.M.G. conceived the project; J.C. and J.C.M.G. designed the project; J.C. performed the single-molecule measurements; T.A.B. contributed to the single-molecule measurements; J.C. analyzed data with contributions from D.S.A. and J.C.M.G.; J.C. performed biochemical techniques; T.A.B. performed phenotype analysis with contributions from J.C.; T.K. and T.V. set up the lattice light-sheet microscope; V.V.M. and A. P. performed bioinformatics analysis; P.I.K. and L.G. performed computational modeling and analysis; J.C., P.I.K., and J.C.M.G. wrote the manuscript with comments from all authors.

## Funding

## Competing interests

The authors declare no competing interests.
