## [Transparent Peer Review file · Nature Communications]

Increasingly efficient chromatin binding of cohesin and CTCF supports chromatin architecture formation during zebrafish embryogenesis

Corresponding Author: Professor J. Christof Gebhardt

Version 0:

Reviewer comments:

Reviewer #1

(Remarks to the Author)

This interesting paper uses tracking of single halo-tagged cohesin (Rad21) or CTCF molecules to examine the bound fraction of cohesin and CTCF in live zebrafish embryos during early embryogenesis. Sophisticated microscopy and computational techniques are employed to try and show that the observations of single molecule movements line up with cohesin binding and loop extrusion. The results are consistent with increasing processivity and formation of genome structure after zygotic genome activation. The manuscript conclusions are largely in line with previous observations of genome structure using HiC methods, but are nonetheless interesting because microscopy methods tracking cohesin haven't been applied to live zebrafish embryos previously.

However, this work appears to be a bit descriptive and is let down by vague language for e.g., in the abstract "Our findings suggest a kinetic framework of chromatin architecture formation during zebrafish embryogenesis." - what does this mean? Overall I am not convinced there is novel mechanistic insight.

Disclaimer for this review: I am a biologist and not a computational scientist. While the methods used to analyse the microscopy data are well described, I am not familiar with them. Some of the paper describes computational polymer modelling, and this is outside my scope of expertise. Therefore here I comment mostly on molecular methods and the biological manipulations of zebrafish, with which there are a number of problems that need to be addressed.

Major issues

Conclusions not adequately supported by evidence, and the methodology needs to be better controlled.

1. Long-bound versus short bound HT-Rad21 and HT-CTCF. The authors assume that long bound HT-Rad21 is incorporated in cohesin. This needs proof, such as co-immunoprecipitation of other cohesin subunits with HT-Rad21. What does it mean that long-bound is associated with the "default chromatin function" of these molecules? There is no independent evidence for this and molecular evidence would be required to confirm.

2. Manipulation of Nipbl and Wapl using morpholinos is very poorly controlled if at all. Morpholinos are good tools but can have off-target and toxicity effects, and their on-target effects need to be tightly controlled. A Western blot is the only evidence supplied of Wapl knockdown, and this blot is unconvincing. An extra lower band appears (or is stronger) in the KD, what is this band? One of the Wapls? Are Wapla and Waplb the same size? What is the real Wapl band? There are many. The quantitation is via densitometry against a saturated tubulin band, and there isn't any evidence that the Wapl antibody specifically cross-reacts with zebrafish as the proteins aren't identified, and mammalian protein is not run as a comparison. The Nipbl KD is not controlled at all, use of reagents just relies on previous publication - there's no independent evidence that these MOs are working in their hands, or to what extent the MOs can deplete Nipbl in early embryos, keeping in mind there will be a high maternal protein load. Several measures could have been used to confirm the reagents are working as predicted, including determining if phenotypes are equivalent to those previously observed, checking with a Rad21 ChIP-PCR etc. If there are off-target effects, these likely would have interfered with the subsequent imaging analyses.

3. More on Wapl KD. There are two Wapls, Wapla and Waplb in zebrafish. As far as I know, wapl mutants and wapl MO

knockdowns have not yet been published. The authors opted to simply target both paralogues with MOs at a single dose in equal amounts, without investigating which of the the *wapl* genes are expressed and at what levels. The authors should at the very least determine which *wapl* is the true orthologue or most active at that stage, and what their relative expression levels are at the stage being targeted. Then to evaluate if the knockdown is working they need independent phenotyping measures, for example, the presence of vermicelli chromosomes in a *wapl* KD.

4. There is a marked switch in the format of presentation of data from fig 1 g,h to fig 1i. The signal behaviour over time is graphs abruptly transition to the snapshot at shield stage once the morphants and chemical treatments are analysed, rather than the graphs that show log bound fraction over time. The authors show the log bound fraction over time in the presence of control morpholinos as binding events over time/stage so why not the actual knock downs? Even if in supplementary data, this information would be good to see so that the data can be compared directly. The omission of these log bound fraction over time graphs makes one suspicious that the data up to shield stage may not be convincing. And if toxicity effects or delays were to manifest as a result of MO treatment, then this could be apparent by shield stage and affect the results. A last word on MOs - it is very difficult if not impossible to target maternally deposited RNA with MOs and completely impossible to target maternally deposited protein, so I'd be very surprised if any of the MOs had much effect pre-shield anyway. Any sort of control proving knock down by other biological or molecular readouts would lend more confidence to the data.

5. The authors suggest they have evidence showing that loop extrusion by cohesin reduces chromatin mobility, as measured by jump distances which get smaller through development. But could the restriction in chromatin movement just be because the cells and therefore nuclei are getting smaller through the cleavage period and development? Could the *Nipbl* KD effect represent developmental delay such that these embryos haven't been through as many cell divisions? These variables are not controlled for, for example a ChIP-seq or cut&run experiment with cohesin in *Nipbl* KD vs control and a cell cycle analysis of cell numbers (so that mobility doesn't depend on the number or size of cells) would be required to validate these claims. The effectiveness of MOs at these early stages where maternal protein exists needs independent validation, as outlined above. The statement "TACO measurements indicate a role of long-bound, presumably loop-extruding cohesin molecules in reducing chromatin mobility." reflects correlation and there's not good evidence for causation via loop-extruding cohesin.

Minor comments

1. Rad21 protein - first letter should be capitalised when referring to the zebrafish protein.
2. Line 65 and fig S1a&b, give the predicted sizes for the HT-Rad21 and HT-CTCF.
3. Fig S1a&b, please make the bands point directly to the proteins.
4. Fig. S2 - what are the predicted sizes of *Wapla* and *Waplb* in zebrafish? They are unlikely to be the same size if both expressed (going by Ensembl). The antibody used was raised against the human protein which seems to be about 150 kD. What is the lower band that increases in intensity in the KD? Are you knocking down one *Wapl* leading to upregulation of the other?
5. Line 587 - "The amount of molecules Nmol at different developmental stages of zebrafish embryos were estimated from published Western blots" - Hard to imagine that estimation from western blots of other labs would be in any way accurate.

(Remarks on code availability)

Not familiar with computational biology. I did look up the *Wapl* genes and the antibody reagents though :)

Reviewer #2

(Remarks to the Author)

In this work, Coßmann and colleagues characterized the chromatin binding properties of the architectural proteins cohesin and CTCF in vivo during zebrafish early development. For this, they followed a single-molecule tracking approach and polymer simulations to calculate the dynamics of the chromatin-bound fractions of rad21 and CTCF, their association-dissociation kinetics and their effect on the establishment of chromatin architecture during early developmental stages, before and after the zygotic genome activation. They nicely show an increase in the fraction of cohesin and CTCF bound to chromatin during development, a progressive restriction of chromatin movement likely due to increased loop extrusion and the decreased association times and increased residence times of both proteins through development. Finally, they simulate the emergence of chromatin structure during development following their empirically determined parameters and compared with real data from embryos, finding a certain degree of similarity.

The manuscript is scientifically sound and well written, the figures are clear and the experiments and analyses are of high quality. Their observations are important for the field, since they provide a molecular basis that explains, at least in part, the progressive emergence of chromatin 3D architecture during the early stages of embryonic development. The in vivo approach is one of the most notable aspects of the work. However, I have some concerns that the authors should address before publication:

a) Major points:

- 1- Regarding the use of injected HT-rad21 and HT-CTCF fusion proteins, did this overexpression cause any visible

developmental phenotype/delay? Do the tagged proteins compete with endogenous proteins for chromatin binding? ChIP-seq experiments in shield stage comparing anti-rad21/CTCF with anti-HT antibodies in injected embryos should provide some insight into this.

2- The overexpression of HT-rad21 and HT-CTCF is estimated to cause an increase of 10% and 25% in rad21 and CTCF protein expression, respectively. However, there is no quantification in the western blots of Suppl. Fig. 1a-b, and its figure legend states that the injection amounts were increased 10-fold compared to the single-molecule measurements to enhance band clarity. Therefore, how did the authors calculate such 10% and 25% increase in protein expression? Also, CTCF protein is maternally provided in zebrafish embryos (see PMID 34518536); is rad21 also maternally provided?

3- The mutant rad21 and CTCF proteins used here reproduce previously reported mutations, but have you tested their effect on DNA binding in zebrafish embryos?

4- In Fig. 1i, it is intriguing that inhibition of RNAPII elongation reduces both the long and short-bound fractions of rad21; if the long-bound fraction is interpreted as DNA bound molecules and the short-bound fraction as unspecific transient interactions, how do you explain that short-bound molecules also increase upon transcriptional elongation inhibition?

5- Regarding the polymer models simulating the effect of cohesin and CTCF on chromatin architecture, the results of the simulation show an establishment of this structure higher than the real HiC data. For instance, Fig. 4b shows a higher increase in contact probability at shorter distances during development than the increase observed in data from Wike et al (green arrowheads). Moreover, this effect is more notable in Fig. 4g, where one can see that insulation around CTCF sites in shield stage is much more prominent for the simulation than in the real HiC. How do you explain these differences? Are there any other factors that counteract the formation of chromatin structure in vivo? Are you considering all CTCF sites for calculating insulation or only those falling within TAD boundaries (if they can be calculated in shield stage)?

b) Minor points:

1- Please, increase the size of Figure 1i.

2- HT-CTCF data is missing from Suppl. Fig. 5a for comparison with the HT control.

3- In Fig. 2b, short and long bound HT-rad21 molecules should be in the same graph and the statistics for their comparison indicated as referred in the text.

4- Please, reorder the labeling of panels in Fig 4 according to the order in which they are cited in the text.

Thank you.

(Remarks on code availability)

Reviewer #3

(Remarks to the Author)

This paper by Cossmann et al. addresses an interesting question regarding the dynamics of CTCF and cohesin association with chromatin during zebrafish development. CTCF and cohesin play key roles in organizing genomes in 3D as well as in repair and sister chromatid cohesion and are of high current interest. To my knowledge, how the chromatin association dynamics of these proteins change during embryo development has not been studied before this paper, making the contribution novel. Moreover, the methodological approaches are quite cutting-edge combined single-molecule imaging in live zebrafish embryos, what appears to be thoughtful analysis of the single-molecule data and also simulations.

Therefore, I think this paper could be of general interest to the wide readership of Nature Communications and a nice contribution to the literature. However, I do have one quite major concern about the extremely low DNA-bound fraction of cohesin and the lack of consideration of sister-chromatid cohesion, as well as several other comments and concerns, I would like to see the authors address. If the authors can fully address these points, I think it could be a nice paper and contribution to the literature.

MAJOR CONCERN: Sister chromatid-cohesion and cohesive cohesin

Cohesin has several functions, including 1) sister-chromatid cohesion and 2) loop extrusion (and others like repair etc.). In this paper, the authors exclusively interpret their results in terms of loop extrusion, but cohesion is required for nuclear division and arguably the most important function of cohesin.

I am concerned about the extremely low bound fraction of HT-rad21. If I am reading Supp Table S2 correctly, the long bound fraction at 128-cell stage is 0.0002.

Zebrafish has 25 chromosomes, so 50 diploid chromosomes. Let's assume we need at least 100 cohesive cohesins per chromosome for proper chromosome segregation at nuclear division, this means we need at least 5000 long-bound cohesins (cohesive cohesin is known to be "infinitely stable", Gerlich 2006). If the long-bound fraction is 0.0002, this means

there needs to be $5000/0.0002 = 25$ million cohesin complexes per cell at least which is completely unreasonable. Even if we assume the long-bound fraction is only 0.003, we still need 1.67 million cohesins per nucleus. Can the authors please estimate the per-cell abundance of cohesin?

I think a plausible explanation for how their observations fit with the known essential cohesion function of cohesin is required.

Also, what is the time-scale of nuclear division throughout the zebrafish developmental stages studied here? I am also concerned about the 100-second reported residence time for cohesin, since cohesive cohesin is known to be extremely stable (Gerlich 2006) as opposed to loop extruding cohesin which has residence times of 15-30 min in mammalian systems.

Beyond making the numbers match, I'd also like to ask the authors to go through the text and interpret their cohesin results both through the lens of cohesions and extrusion, instead of exclusively focusing on extrusion.

MAJOR COMMENTS

1. PROTEIN LEVELS: line 65 says HT-rad21 and HT-CTCF increases the total rad21 and TCF abundance by 10% and 25% respectively. But when I look at the Westerns in Fig. S1a,b it looks like the HT-rad21 band is about same as wt-rad21 (so 100% increase) and when I look at HT-CTCF it looks like at least 5x (so 500% increase). Can the authors please include careful quantification (including replicate-to-replicate error bars (similar to Fig. S1d)). Getting these numbers right is crucial for interpretations of the rest of the paper.
2. IS RAD21 FUNCTIONAL? Along the lines of #1, when you over-express a sub-unit of a multi-protein complex, the over-expressed sub-unit may not be functional. It seems the authors did N-terminal HT tag on rad21 which is unusual (most papers do C-term). How did the authors validate the HT-rad21 is functional and incorporated in cohesin complexes along with the other cohesin subunits?
3. PROVIDE DATA? I appreciate the authors making the raw data available, but when I clicked on https://datadryad.org/stash/share/zf4_oJgrWv-741_Xb_oSMaNTmYbHZhe5FDHdPz6QRumtil I get a "403 Forbidden" error message. Maybe the provided URL is incorrect, or maybe I am doing something incorrect. Could you provide a working URL to the data?
4. SECTION LINE 110-129 appears to be missing figure references? Please insert references to the figure panels? For Pol II inhibition, what was the treatment length? Presumably Pol II inhibition is lethal? How long can Pol II be inhibited before the animal dies or serious artifacts occur?
5. LINE 145: The NIPBL perturbation is interesting, but it is exclusively interpreted in terms of extrusion. Schwarzer...Spitz Nature 2017 showed that del-NIPBL removes all cohesin from chromatin, consistent with NIPBL loading cohesin. How much does the DNA-bound population of HT-rad21 change? Can you please compare to Schwarzer 2017?
6. LINE 210-245 SIMULATIONS: The simulations are interesting, but reported in an unconventional manner (association rates). To make it easier to compare with other loop extrusion polymer simulation studies can the authors also report it in the standard way: extruder processivity, density/separation, speed (they do report speed). Can they please include the best fit of these numbers in main Fig 4 and in the main text? Can they also please compare to mammalian systems? The arrowheads in Fig. 4b,c,e,f would benefit from having the x-axis value listed (e.g. ~300kb).
7. In the Movies, which look really nice by the way, why are there so frequently molecules outside the nucleus?

SMALLER MINOR COMMENTS

- i. Line 41: along with Ref 6,25, authors should also cite de Wit Mol Cell 2015 for CTCF polarity.
- ii. Really small thing, but why is rad21 all lower case but CTCF all upper case throughout the paper?
- iii. Fig 1d,e shows a lot of data, but really difficult to see by eye whether a fraction is 0.03 or 0.06 because it is so zoomed out. Can the authors start the y-axis at 0.01 or show a zoom-in inset or something along those lines so we can see the values?
- iv. Line 76-78: Sorry if I missed it, but can you please provide the diffusion coefficients associated with each fraction? You also mention anomalous diffusion – how did you determine that the diffusion is anomalous, what kind (e.g. fraction Brownian motion?) and what alpha did you measure?
- v. Line 112, Authors should probably cite Tedeschi Nature 2013 for WAPL
- vi. Line 150 "increase in chromatin architecture" is a bit vague – at least I do not know what it means. Can you rephrase?
- vii. Line 291, Ref 88 is a bit messy and difficult to interpret since long-term partial Pol II depletion can have all sorts of indirect and secondary effects. Probably Banigan PNAS 2023 is a better reference.

(Remarks on code availability)

The data URL does not work, so although I tried, I could not evaluate since the provided URL is not working.

Version 1:

Reviewer comments:

Reviewer #1

(Remarks to the Author)

I appreciated the authors' detailed response to my comments and those of the other reviewers. I think it was a good call to leave out the Wapl experiments despite their promise, owing to the substantial extra characterisation needed in a first-time knock down model. I agree that the omission of these results does not impact on the overall findings. I'm comfortable that the revised manuscript is robust, and is of significance to the field.

Minor comment:

- could the authors please label the Stag subunit in figure 1a? Confusing if not labelled, one might assume it's a structure like the ATPase heads of the SMC proteins.

Reviewer #2

(Remarks to the Author)

I thank the authors for successfully addressing all my concerns by new experimental data and clarifications. In particular, the subcellular fractionation experiments demonstrating chromatin binding of the HT-Rad21 and HT-CTCF and the loss of this binding in mutant alleles HT-Rad21-3x and HT-CTCF- Δ ZF4-7 are valuable controls to support their conclusions. This also applies to the Co-IP experiments showing the formation of cohesin complexes by HT-Rad21, and to the phenotypic characterization of injected embryos.

I think that this work is an important contribution to the field and that this revised version is very suitable for publication in Nature Communications.

Reviewer #3

(Remarks to the Author)

Cossmann et al. have resubmitted a revised manuscript, and the reviewer response is quite extensive and it is clear the authors have put a lot of work and care into the revisions.

Generally speaking, I found the revisions well-done. There were also a couple of things I missed (seems like I used the wrong URL to access the data due to a line number issue and a couple of things I asked for was already in SI Figures).

As I mentioned in the first review, I generally find the study compelling and innovative, but my major concern is the extremely low bound fraction of cohesin which appears incompatible with the requirement for cohesin in replication. The authors to their credit explicitly discuss this in their Discussion section – and I think including this is absolutely essential. I do find it much more likely that the authors slightly misestimated the bound fraction than I do there being >1 million cohesins per nucleus.

Along these lines the authors write “The number of cohesive cohesin molecules required for sister chromatid cohesin is not expected to change during development.”. Unless “per cell” is included, this statement is incorrect. If the maternally deposited number of cohesin proteins complexes is M and the number of cohesins required per nucleus is N and the cell division number is X, then the number of cohesins required is N^X which will grow rapidly and quickly exceed M. Every time the number of nuclei double during embryogenesis, the required number of cohesins per embryo also double. So, the authors should modify this sentence.

But at the end of the day, if the authors feel comfortable with this conclusion I think the study should get published as long as the critical discussion they included in the revision is also included in the final paper, then readers can decide for themselves.

I also think it would be good for the authors to state that they cannot exclude a subpopulation of very long-lived cohesins missed by the single-molecule tracking, since cohesive cohesins will have lifetimes of at least S- + G2-phase duration, which appears to exceed their currently estimated residence time.

Thus, in conclusion I think this is a technically very impressive study that addresses some very interesting and previously unaddressed questions related to the establishment of 3D chromatin architecture in embryogenesis. I do have concerns about the low bound fractions and short residence times of cohesins, but as long as these are openly and explicitly discussed as the authors do in their current revision, I consider all my questions and concerns addressed and recommend the paper for publication in Nat Comm.

VERY SMALL THINGS:

Line 166 where they cite 101-103 REFs, this was also shown in PMID: 35420890 which should also be cited here.

Response to reviewers' comments

We thank the reviewers for their thoughtful and constructive feedback. Below, we address each point individually. We highlighted additional material and revised text passages in our manuscript in yellow.

Reviewer #1 (Remarks to the Author):

This interesting paper uses tracking of single halo-tagged cohesin (Rad21) or CTCF molecules to examine the bound fraction of cohesin and CTCF in live zebrafish embryos during early embryogenesis. Sophisticated microscopy and computational techniques are employed to try and show that the observations of single molecule movements line up with cohesin binding and loop extrusion. The results are consistent with increasing processivity and formation of genome structure after zygotic genome activation. The manuscript conclusions are largely in line with previous observations of genome structure using HiC methods, but are nonetheless interesting because microscopy methods tracking cohesin haven't been applied to live zebrafish embryos previously.

However, this work appears to be a bit descriptive and is let down by vague language for e.g., in the abstract "Our findings suggest a kinetic framework of chromatin architecture formation during zebrafish embryogenesis." - what does this mean? Overall I am not convinced there is novel mechanistic insight.

We thank the reviewer for highlighting both positive aspects of our study and areas that need clarification.

Our *in vivo* live-embryo kinetic measurements of CTCF and cohesin, our quantitative analysis, and simulations add previously unknown molecular and kinetic parameters such as bound fractions, association, and dissociation rates, and their changes embedded into a physico-chemical framework and the developmental context, thus providing important new mechanistic insight.

We revised our abstract to better summarize our findings: "Our findings reveal molecular kinetics underlying chromatin architecture formation during zebrafish embryogenesis."

Disclaimer for this review: I am a biologist and not a computational scientist. While the methods used to analyse the microscopy data are well described, I am not familiar with them. Some of the paper describes computational polymer modelling, and this is outside my scope of expertise. Therefore here I comment mostly on molecular methods and the biological manipulations of zebrafish, with which there are a number of problems that need to be addressed.

Major issues

Conclusions not adequately supported by evidence, and the methodology needs to be better controlled.

1. Long-bound versus short bound HT-Rad21 and HT-CTCF. The authors assume that long bound HT-Rad21 is incorporated in cohesin. This needs proof, such as co-immunoprecipitation of other cohesin subunits with HT-Rad21.

We agree with the reviewer. As suggested, we now performed co-immunoprecipitation of Smc3 using an antibody against the HT-Rad21 fusion protein. Similar to what we described for reviewer 3 (point 2), we utilized the two 5' located HA-tags in our HT-Rad21 construct, and performed a pull-down on shield-stage protein lysates from zebrafish embryos injected with HT-rad21. Since the low injection amount of 0.67 pg of mRNA encoding for HT-Rad21 we used in single-molecule experiments did not allow for visualization of the HT-Rad21 protein band in Western blots, we injected 10-fold higher amounts than used for single-molecule experiments. We validated the IP with a positive HA-tagged protein (Pierce™ HA-Tag IP/Co-IP Kit, Thermo Fisher Scientific) and uninjected zebrafish embryo lysate without HA-tag as a negative control. To unambiguously identify each protein, we ran equally distributed samples of a single IP on two blots (see below, new Supplementary Fig. 2a, b). In the first blot (Panel a), we verified the presence of the HA-tag in the input and pull-down of HT-rad21-injected, but not the uninjected WT control embryos (for band identification, see Western Blot Supplementary Fig. 1). Using an anti-Smc3 antibody, we identified the presence of Smc3 on the second blot in the Input samples of both HT-rad21-injected embryos and uninjected controls (Panel b). Importantly, Smc3 was only detected in the HA-tag pull-down of the HT-rad21-injected sample but not in the HA-tag pull-down of the uninjected control embryos (Panel b). To visualize both HT-Rad21 and Smc3 proteins on a single blot, we probed the second blot once more with the anti-HA tag antibody (Supplementary Fig. 2c). This indicates that HT-Rad21 is successfully incorporated into the cohesin complex.

Supplementary Figure 2: Co-immunoprecipitation (Co-IP) of Smc3 with HA-HT-Rad21 in zebrafish embryos. SDS-page gel stained with **a)** anti-HA tag antibody (Blot 1) and **b)** anti-Smc3 antibody (Blot 2). **c)** Blot 2 from panel b) subsequently stained with anti-HA tag antibody. Gels were loaded with: lysate of embryos injected with mRNA encoding for HA-tagged HT-Rad21 (lane 1, Input), immunoprecipitation (IP) using an anti-HA tag antibody (lane 2, IP HA), lysate of uninjected embryos (lane 3, WT, Input), IP using an anti-HA tag antibody (lane 4, IP HA). Endogenous Smc3 was detected in both the HA-HT-rad21 injected and uninjected WT input samples. Smc3 was also detected in the IP HA samples from injected embryos. HA-HT-Rad21 protein was only detected in embryos injected with the HA-HT-rad21 mRNA and not in the WT control.

We included our observation in the main manuscript: "We confirmed the incorporation of HT-Rad21 into the cohesin complex through co-immunoprecipitation of the N-terminally interacting Smc3 protein¹ (Supplementary Fig. 2)". We further added Supplementary Figure 2 and described the Co-IP in the methods section.

What does it mean that long-bound is associated with the "default chromatin function" of these molecules? There is no independent evidence for this and molecular evidence would be required to confirm.

We admit that this was not clearly formulated. To confirm that HT-Rad21 and HT-CTCF indeed bind chromatin, we conducted subcellular protein fractionation experiments in shield-stage zebrafish embryos injected in the 1-cell stage. We again injected 10-fold higher HT-rad21 mRNA amounts (6.7 pg) or HT-ctcf mRNA amounts (10 pg) compared to our single-molecule experiments (0.67 pg and 1.0 pg, respectively), to allow for Western blot analysis. All extracts (cytoplasmic (1-CE), membrane (2-ME), soluble nuclear (3-SNE), and cytoskeletal (5-CSE)) were probed using an anti-Rad21 or anti-CTCF antibody. We validated the chromatin bound origin of the chromatin-bound extract (4-CBE) by the presence of Histone H2B. In addition to the expected endogenous protein bands, we observed HT-Rad21 and HT-CTCF bands in all extracts (see Supplementary Fig. 2 below, and Western blots quantifying HT-Rad21 overexpression (Supplementary Fig. 1) for band identification). Of note, the relative amounts of HT-Rad21 and HT-CTCF in each fraction closely recapitulate the distributions of endogenous Rad21 and CTCF. From these assays, we conclude that HT-Rad21 and HT-CTCF are indeed able to associate with chromatin to a similar extent as the endogenous proteins. Since Rad21 binds to chromatin together with Smc1/3²⁻⁵, the subcellular protein fractionation strengthens our conclusion from the CoIP assay that HT-Rad21 is able to incorporate into the cohesin complex.

Supplementary Figure 3. Subcellular protein fractionation of HT-tagged wild-type proteins from lysates of shield-stage zebrafish embryos. a)-b) Western blots of subcellular protein fractionated shield-stage embryos injected in the 1-cell stage with a) 6.7 pg HT-rad21 mRNA or b) 10 pg HT-ctcf mRNA. Blots were probed with anti-Rad21 and anti-Histone H2B antibody (a), or anti-CTCF and anti-Histone H2B antibody (b). Embryos were injected with a 10-fold injection amount compared to our single-molecule measurements to enhance band clarity (see Methods). Lanes include extracts of: 1-CE: Cytoplasmic, 2-ME: Membrane, 3-SNE: Soluble nuclear, 4-CBE: Chromatin bound, 5-CSE: Cytoskeletal extracts.

We repeated the protein fractionation assay for HT-Rad21-3x (Supplementary Fig. 12a below). This revealed a strong reduction of the chromatin-bound fraction of the HT-Rad21-3x mutant compared to HT-Rad21-WT, normalized to endogenous Rad21. Thus, we conclude that chromatin binding of HT-Rad21-3x is indeed impaired, as reported^{2,3,5}. Similarly, the binding-deficient mutant HT-CTCF-ΔZF4-7 did not show chromatin binding (Supplementary Fig. 12b, c below; for band identification, see Western blot Supplementary Fig. 1).

Supplementary Figure 12. Subcellular protein fractionation of HT-tagged mutant proteins from lysates of shield-stage zebrafish embryos. a)-b) Western blots of subcellular protein fractionated shield-stage embryos injected in the 1-cell stage with a) 6.7 pg HT-rad2- 3x mutant mRNA or b) 10 pg HT-ctcf-ΔZF4-7 mutant mRNA. Blots were probed with anti-Rad21 and anti-Histone H2B antibody (a)), or anti-CTCF and anti-Histone H2B antibody (b)). c) Intensity plot for Western blot bands depicted in panel b) between ~180-130 kDa (refer to grey marker lane), quantified across a straight region of each band (see Methods). Asterisk denotes the intensity maximum corresponding to HT-CTCF-ΔZF4-7 protein. Embryos were injected with a 10-fold injection amount compared to our single-molecule measurements to enhance band clarity (see Methods). Lanes include extracts of: 1-CE: Cytoplasmic, 2-ME: Membrane, 3-SNE: Soluble nuclear, 4-CBE: Chromatin bound, 5-CSE: Cytoskeletal extracts.

Overall, the results from the chromatin fractionation assays confirmed chromatin binding of HT-Rad21 and HT-CTCF and impaired chromatin binding of HT-Rad21-3x and HT-CTCF-ΔZF4-7. Together with our observation that impaired chromatin binding affects the long-bound fraction but not the short-bound fraction of HT-Rad21 and HT-CTCF, respectively, this indicates that long-binding events represent a chromatin binding mode of wild-type proteins, while short-bound proteins represent other modes of binding.

We added the results of the protein fractionation assays in the main text: "We further confirmed chromatin association of HT-Rad21 and HT-CTCF by subcellular protein fractionation (Supplementary Fig. 3)." and "We confirmed impaired chromatin binding of HT-Rad21-3x and HT-CTCF-ΔZF4-7 by subcellular protein fractionation (Supplementary Fig. 12)." We further added Supplementary Fig. 3 and 12, and a methodological description in the methods section.

In the results and discussion, we now try to express our findings more precisely: "Overall, our ITM results indicate that the long-bound fractions of HT-Rad21 or HT-CTCF are predominantly associated with chromatin binding, while short-bound fractions are presumably associated with unspecific transient modes of binding to DNA, RNA, proteins or other biomolecules, or transient spatial confinement."

2. Manipulation of Nipbl and Wapl using morpholinos is very poorly controlled if at all. Morpholinos are good tools but can have off-target and toxicity effects, and their on-target effects need to be tightly controlled. A Western blot is the only evidence supplied of Wapl knockdown, and this blot is unconvincing. An extra lower band appears (or is stronger) in the KD, what is this band? One of the Wapls? Are Wapla and Waplb the same size? What is the real Wapl band? There are many. The quantitation is via densitometry against a saturated tubulin band, and there isn't any evidence that

the Wapl antibody specifically cross-reacts with zebrafish as the proteins aren't identified, and mammalian protein is not run as a comparison.

We appreciate the reviewer's concerns regarding the specificity of the anti-human WAPL antibody against zebrafish proteins in Western blots and we agree that additional controls are important.

To validate our commercial human wapl antibody against zebrafish Wapla and Waplb, we isolated the corresponding wapl genes from zebrafish dome stage cDNA, in vitro synthesized (IVS) the Wapla and Waplb proteins, and performed Western blot analysis using our anti-Wapl antibody (Figure on the right). Our results show that the human anti-Wapl antibody successfully detects both *D. rerio* Wapla and Waplb proteins. Due to the similar molecular weight, we cannot distinguish between Wapla and Waplb paralogs with this antibody. Importantly, this experiment enables us to unambiguously identify the Wapla/b band in Western blots of wild-type (WT) and wapl-MO injected embryos.

In-Vitro synthesized (IVS) D. rerio Wapla and Waplb protein with anti-Wapl antibody.

We further extended quantifying the reduction of Wapl protein in shield stage upon injection of both wapl-MOs into 1-cell stage embryos (Figure on the right), and found a reduction of Wapl protein down to 20% with 4 ng of injected MO.

Quantification of Wapl protein levels from Wapl-MO injected embryos compared to WT and loading control from Western blots.

Although these additional controls of our wapl-MOs are very promising, they are not sufficient to establish a new MO, according to published guidelines⁶. We decided to abandon further rigorous validation of these MOs and further controls of corresponding measurements due to time constraints in the project. Our results of HT-Rad21 bound fraction upon reduction of Wapl protein are not relevant for our conclusions and do not add to our kinetic model. We thus omitted the measurements with wapl-MO from the manuscript and instead focused on controls and analysis critical to the conclusions of the manuscript.

The Nipbl KD is not controlled at all, use of reagents just relies on previous publication - there's no independent evidence that these MOs are working in their hands, or to what extent the MOs can deplete Nipbl in early embryos, keeping in mind there will be a high maternal protein load. Several measures could have been used to confirm the reagents are working as predicted, including determining if phenotypes are equivalent to those previously observed, checking with a Rad21 ChIP-PCR etc. If there are off-target effects, these likely would have interfered with the subsequent imaging analyses.

As suggested, we now compared embryo phenotypes with the original publication of the nipbl-MOs. We tested embryos injected with nipbla-MO, nipblb-MO, both MOs in combination, a standard-control morpholino, and a non-injected wild-type (WT) control (see below, Supplementary Fig. 16). We monitored embryonic development up to 48 hpf. Consistent with the original publication, we did not observe significant developmental differences up to shield stage, the stage in which we performed our measurements. Developmental defects became apparent at 24 hours post fertilization (hpf) with a shortened tail. Phenotypes were more pronounced at 48 hpf, especially for nipblb2-MO and the

coinjection of nipbla1- and nipblb2-MO. Similar to the original publication⁷, we observed pericardial edema and tail defects, plus developmental delays and non-specific proliferation defects up to necrosis. Since neither Nipbla nor Nipblb antibodies are commercially available, we did not include Western blot analysis of the knockdown efficiency.

Supplementary Figure 16:
Development of nipbl-morpholino(MO) injected zebrafish embryos. Lateral views of whole embryos either uninjected or injected with standard control MO (Gene Tools LLC, USA), nipbla1-MO, nipblb2-MO, nipbla1- and nipblb2-MO (see Methods). At 24 hours post-fertilization (hpf), nipblb2-MO and coinjected nipbla1/nipblb2-MO zebrafish exhibit shortened tails. By 48 hpf, each nipbl-MO and the coinjection show pericardial edema and tail defects. Coinjection additionally causes developmental defects and non-specific proliferation defects at 48 hpf. Images of coinjected fish at 48 hpf exemplify a range of phenotypes, including tail and proliferation defects up to necrosis. Stages are provided according to Kimmel et al., 1995. Scale bar is 500 μ m and applies to all images.

For the ctcf-MO, we now performed a Western blot analysis (see below, Supplementary Fig. 17) to assess knockdown efficiency in shield stage, showing a 6.4-fold reduction of CTCF protein to 16 % of the WT expression level upon MO injection (Panel b).

Supplementary Figure 17: Quantification of CTCF expression level after *ctcf*-morpholino (MO) injection. *a*) Western blots with anti-CTCF and anti- γ -Tubulin (γ -Tub.) antibodies of shield-stage zebrafish embryo lysate after injection of *ctcf*-MO. *b*) Quantification of CTCF protein level from *ctcf*-MO injected embryos compared to WT and loading control shown in panel a). Lines represent mean value \pm s.d.

In addition, we monitored phenotypes at shield-stage, at 24 hpf, and 48 hpf after *ctcf*-MO injection and compared with the original publication (see below, Supplementary Fig. 18). We observed a developmental delay already at shield stage, similar to the original publication⁸, and lethality at 24 hpf. CTCF depletion is expected to be lethal^{9,10}.

Supplementary Figure 18: Development of *ctcf*-morpholino (MO) injected zebrafish embryos. Lateral views of whole embryos either uninjected or injected with standard control MO (Gene Tools LLC, USA) or *ctcf*-MO (see Methods). Embryos injected with *ctcf*-MO exhibited developmental delays, remaining at around 50% epiboly, compared to the time point of shield stage in both uninjected and standard control-MO injected embryos. Injection of *ctcf*-MO was lethal beyond 24 hours post-fertilization (hpf). Stages are provided according to Kimmel et al., 1995. Scale bar is 500 μ m and applies to all images.

We included our observations in the main manuscript: “We confirmed normal development of *nipbl*-MO injected embryos until shield-stage and severe developmental defects at 48 hpf (Supplementary Fig. 16), as reported previously⁷. Upon injection of *ctcf*-MO, the expression level of endogenous CTCF protein was reduced down to $(15.6 \pm 2.0)\%$ at the time point of shield stage in untreated embryos (Supplementary Fig. 17). Moreover, we confirmed a developmental delay in shield-stage and lethality at later stages in *ctcf*-MO injected embryos, as previously reported (Supplementary Fig. 18)^{9,10}. “. We also included Supplementary Figures 17-18 and a methods section describing the *ctcf*-MO Western blots in the manuscript.

3. More on Wapl KD. There are two Wapls, Wapla and Waplb in zebrafish. As far as I know, wapl mutants and wapl MO knockdowns have not yet been published. The authors opted to simply target both paralogues with MOs at a single dose in equal amounts, without investigating which of the the wapl genes are expressed and at what levels. The authors should at the very least determine which wapl is the true orthologue or most active at that stage, and what their relative expression levels are at the stage being targeted. Then to evaluate if the knockdown is working they need independent phenotyping measures, for example, the presence of vermicelli chromosomes in a wapl KD.

We agree with the reviewer that a more rigorous characterization of the wapl-MOs would be necessary. However, due to a lack of commercially available antibodies specific to individual wapl paralogs, and the comparable size of both paralogs, characterization as suggested by the reviewer was not possible. As we mentioned in our answer to point 2, we now omitted our measurements using the wapl-MOs from the manuscript. Importantly, these measurements were not relevant to our conclusions and did not add to our kinetic model. Omitting the wapl-MO measurements therefore do not affect the remainder of the manuscript.

4. There is a marked switch in the format of presentation of data from Fig 1 g,h to fig1i. The signal behaviour over time is graphs abruptly transition to the snapshot at shield stage once the morphants and chemical treatments are analysed, rather than the graphs that show log bound fraction over time. The authors show the log bound fraction over time in the presence of control morpholinos as binding events over time/stage so why not the actual knock downs? Even if in supplementary data, this information would be good to see so that the data can be compared directly. The omission of these log bound fraction over time graphs makes one suspicious that the data up to shield stage may not be convincing. And if toxicity effects or delays were to manifest as a result of MO treatment, then this could be apparent by shield stage and affect the results.

We understand the concern of the reviewer. Our intention for Fig. 1i is to focus on the changes between measurement conditions rather than the changes between stages, as in Fig. 1g,h. Moreover, comparing small numbers, such as the bound fractions observed at early stages, is intrinsically more error-prone than comparing larger values as observed in shield stage (base effect of low numbers). Therefore, we opted to display only shield-stage data, where bound fractions are highest, to make this comparison most robust. In addition, as also pointed out by the reviewer, treatment with MOs is expected to be ineffective at pre-ZGA stages due to the maternal supply of proteins, making interpretation of MO data questionable at early stages.

Nevertheless, as suggested, we now also show in the Supplementary Information the changes in bound fractions between developmental stages for nipbl-MO (see below and Supplementary Fig.19a, Supplementary Tables 21 and 22), ctcf-MO (Supplementary Fig. 19b, Supplementary Tables 21 and 22), α -Amanitin (Supplementary Fig. 22a, Supplementary Tables 23 and 24) and Triptolide (Supplementary Fig. 22b, Supplementary Tables 23 and 24). Except for ctcf-MO injected and α -Amanitin-treated embryos, the bound fractions mostly do not deviate from untreated embryos. For ctcf-MO and α -Amanitin, we observe significant deviations for several successive stages in a row, but cannot exclude that these are due to the base effect of low numbers. Therefore, we do not want to overinterpret these changes.

We now provide these additional data together with statistics and p-Values in the Supplementary Figures 19, 22, and Supplementary Tables 21-24.

Supplementary Figure 19: Short and long bound fractions of HT-Rad21 in presence of nipbl-morpholino (MO), ctf-MO, or standard control MO. Fractions of short and long binding events from ITM measurements of HT-Rad21 in **a**) embryos injected with nipbl-MO (HT-Rad21 + nipbl-MO) or uninjected embryos (HT-Rad21), **b**) embryos injected with ctf-MO (HT-Rad21 + ctf-MO) or uninjected embryos (HT-Rad21) and **c**) embryos injected with standard control-MO (HT-Rad21 + standard control-MO) or uninjected embryos (HT-Rad21). Data represent mean \pm s.d. of movie-wise determined fractions. Insets show zooms into the respective graphs. Lines serve as guides to the eye.

Supplementary Figure 22: Short and long bound fractions of HT-Rad21 in presence of α -Amanitin or Triptolide. Fractions of short and long binding events from ITM measurements of HT-Rad21 in **a**) embryos treated with α -Amanitin (HT-Rad21 + α -Amanitin) or untreated embryos (HT-Rad21) and **b**) embryos treated with Triptolide (HT-Rad21 + Triptolide) or untreated embryos (HT-Rad21). Data represent mean \pm s.d. of movie-wise determined fractions. Insets show zooms into the respective graphs. Lines serve as guides to the eye.

A last word on MOs - it is very difficult if not impossible to target maternally deposited RNA with MOs and completely impossible to target maternally deposited protein, so I'd be very surprised if any of the MOs had much effect pre-shield anyway. Any sort of control proving knock down by other biological or molecular readouts would lend more confidence to the data.

We agree. Therefore, we do not intend to interpret the bound fractions of HT-Rad21 in MO-treated embryos prior to shield stage. To address the reviewers concerns, we now added additional controls to show functioning of the established MOs in our lab: characterization of phenotypes for nipbl-MO injected embryos (Supplementary Fig. 16), quantification of CTCF expression level in shield stage via Western blots (Supplementary Fig. 17), and phenotypes for ctf-MO injected embryos (Supplementary Fig. 18). In addition, we quantified developmental delays at early stages (see next Point 5 and Supplementary Fig. 4).

5. The authors suggest they have evidence showing that loop extrusion by cohesin reduces chromatin mobility, as measured by jump distances which get smaller through development. But could the restriction in chromatin movement just be because the cells and therefore nuclei are getting smaller through the cleavage period and development? Could the Nipbl KD effect represent developmental delay such that these embryos haven't been through as many cell divisions? These variables are not controlled for, for example a CHIP-seq or cut&run experiment with cohesin in Nipbl KD vs control and a cell cycle analysis of cell numbers (so that mobility doesn't depend on the number or size of cells) would be required to validate these claims. The effectiveness of MOs at these early stages where maternal protein exists needs independent validation, as outlined above. The statement "TACO measurements indicate a role of long-bound, presumably loop-extruding cohesin molecules in

reducing chromatin mobility." reflects correlation and there's not good evidence for causation via loop-extruding cohesin.

We agree with the reviewer that differences in the observed jump-distances in the absence or presence of nipbl-MO might be the result of both developmental delay or less bound cohesin upon Nipbl knockdown.

As suggested, we now controlled for a possible developmental delay at early stages by analyzing the time gap between stages using the time-stamps of our single-molecule movies (see Supplementary Fig. 4, below). Since we co-injected GFP labeled Lap2 β , a nuclear membrane marker, we could precisely track cell divisions during measurements for various conditions. For the nipbl-MO injected embryos (yellow star) we did not observe a consistent developmental delay in the early measurements compared to HT-Rad21 (dark green triangle), the standard control morpholino (purple circle), or HaloTag control (pink circle). For shield stage we refer to our phenotype characterization, where we could not observe a developmental delay between HT-Rad21 or HT-Rad21+nipbl-MO injected embryos up to shield stage, consistent with the original publication of nipbl-MO (Supplementary Fig. 16)⁷.

Supplementary Figure 4. Time intervals between developmental stages of early zebrafish embryos injected with different mRNAs or treatments. Time between developmental stages was determined by comparing time-stamps of consecutive raw movies for each embryo (see Methods). Data represent mean values \pm s.d.

Together, these experiments ensure that nipbl-MO injection does not lead to a developmental delay compared to control embryos up to shield stage, were we compared the jump-distances. We included the Supplementary Figures 4 and 16, a methods section detailing the time interval determination and references in the main manuscript.

Moreover, the reviewer is absolutely right that also an overall lower amount of cohesin bound to chromatin in presence of nipbl-MO might contribute to the increased mobility of chromatin, even if we characterized the mobility of cohesin-bound loci. We have so far overlooked this possibility since we characterized the mobility of local cohesin-bound loci. Indeed, multiple publications demonstrated a reduced association of cohesin to chromatin upon lower Nipbl levels by ChIP-seq¹¹⁻¹⁵. We thank the reviewer for pointing out this additional mechanism of increasing chromatin mobility, which we now highlight in the manuscript as a further possibility besides a lower Nipbl-mediated ATPase activity: "Since Nipbl both reduces chromatin-bound cohesin¹¹⁻¹⁵ and stimulates the ATPase activity of

cohesin¹⁶⁻¹⁹, both a reduced cohesive function and reduced loop extrusion activity might contribute to increased chromatin mobility”.

We further extended our TACO measurements to include mean jump distances of short bound HT-Rad21 tracks under nipbl-MO treatment in shield-stage (see panel b of Fig. 2 below). If a developmental delay were responsible for the increase in mean jump distances of long-bound tracks, a similar increase should occur in the mean jump distances of short-bound tracks. However, we did not observe a significant increase in the mean jump distances of short-bound tracks under nipbl-MO treatment compared to the reference HT-Rad21 measurement.

We included this important control in Fig. 2b of the main manuscript and statistics in Supplementary Tables 10-11.

Figure 2b: Mean jump distances of HT-Rad21 molecules classified as short bound (circles) and long bound (squares) at different developmental phases. pre-ZGA (red): 64-, 128-, 256, 512-cell stages pooled; post-ZGA (green): high, oblong, sphere stages pooled; shield stage (blue) and shield stage with the addition of nipbl-morpholino (MO, cyan). Insets: example tracks. Data represent mean \pm s.d. P-values were calculated using the Kruskal-Wallis-Test and Multiple Mann-Whitney-Test. Scale bar: 1 μ m.

Minor comments

1. Rad21 protein - first letter should be capitalised when referring to the zebrafish protein.

Thank you for pointing this out. We have corrected this where necessary.

2. Line 65 and fig S1a&b, give the predicted sizes for the HT-Rad21 and HT-CTCF.

As suggested, we have added the predicted sizes for our HT-Rad21 (109.2 kDa) and HT-CTCF (126.8 kDa).

3. Fig S1a&b, please make the bands point directly to the proteins.

We now show the relevant bands of replicate Western blots at the same height to facilitate comparison. We further changed the arrows to have horizontal lines to better clarify the height they

point to. Moreover, we added the predicted molecular weight of proteins across all Western blots. We hope these measures facilitate the readability of our data.

4. Fig. S2 - what are the predicted sizes of Wapla and Waplb in zebrafish? They are unlikely to be the same size if both expressed (going by Ensembl). The antibody used was raised against the human protein which seems to be about 150 kD. What is the lower band that increases in intensity in the KD? Are you knocking down one Wapl leading to upregulation of the other?

The molecular sizes of Wapl proteins are reported to be of similar size for Zebrafish Wapla (127,130 Da), Zebrafish Waplb (129,733 Da), Human WAPL (132,946 Da), and Mouse WAPL (134,070 Da). We demonstrated recognition of in vitro synthesized Wapla and Waplb by our antibody in Western blots (described above in comment 2), which enabled us to infer the molecular weight and running height of the Zebrafish Wapl proteins. This had allowed us to unambiguously identify the full-length Wapl protein in our Western blots. Since lower molecular weight bands also occurred in control samples, we hypothesized that they represent truncated protein. As explained above, we now removed the experiments with wapl-MO from the manuscript.

5. Line 587 - "The amount of molecules N_{mol} at different developmental stages of zebrafish embryos were estimated from published Western blots" - Hard to imagine that estimation from western blots of other labs would be in any way accurate.

We apologize for not being clear enough in our description. We did not ourselves quantify the published Western blots but obtained the values from the quantification graphs. Moreover, our intention here was not to arrive at accurate values. Instead, our aim was to get a coarse estimate of the order of magnitude of protein levels. We now clarify this in the manuscript: "Therefore, we included an order-of-magnitude estimate of changes in relative protein concentrations during early zebrafish development determined from published data⁹ and our nuclear volume measurements (Supplementary Fig. 27 and Methods)". We also clarified it in the method section: "The amount of molecules N_{mol} at different developmental stages of zebrafish embryos were approximated from published quantification data of Western blots⁹."

Reviewer #1 (Remarks on code availability):

Not familiar with computational biology. I did look up the Wapl genes and the antibody reagents though :)

We thank the reviewer for their thorough reading of our manuscript and their constructive criticism.

Reviewer #2 (Remarks to the Author):

In this work, Coßmann and colleagues characterized the chromatin binding properties of the architectural proteins cohesin and CTCF in vivo during zebrafish early development. For this, they followed a single-molecule tracking approach and polymer simulations to calculate the dynamics of the chromatin-bound fractions of rad21 and CTCF, their association-dissociation kinetics and their effect on the establishment of chromatin architecture during early developmental stages, before and after the zygotic genome activation. They nicely show an increase in the fraction of cohesin and CTCF bound to chromatin during development, a progressive restriction of chromatin movement likely due to increased loop extrusion and the decreased association times and increased residence times of both proteins through development. Finally, they simulate the emergence of chromatin structure during development following their empirically determined parameters and compared with real data from embryos, finding a certain degree of similarity.

The manuscript is scientifically sound and well written, the figures are clear and the experiments and analyses are of high quality. Their observations are important for the field, since they provide a molecular basis that explains, at least in part, the progressive emergence of chromatin 3D architecture during the early stages of embryonic development. The in vivo approach is one of the most notable aspects of the work. However, I have some concerns that the authors should address before publication:

We thank the reviewer for their detailed and constructive feedback on our manuscript and appreciate their enthusiastic assessment. Below, we carefully address all concerns to improve the quality and clarity of our study.

a) Major points:

1- Regarding the use of injected HT-rad21 and HT-CTCF fusion proteins, did this overexpression cause any visible developmental phenotype/delay?

As suggested, we now characterize potential developmental delays for different experimental conditions in more detail. We performed phenotype analysis for our 0.67 pg HT-rad21 and 1pg HT-ctcf injections at 4 hpf, shield stage, 24 hpf, and 48 hpf (Supplementary Fig. 5, below). For both fusion proteins, we did not observe a developmental delay. We added the graph as Supplementary Fig. 5 and mention the results in the main manuscript: "The HT-rad21 and HT-ctcf injected embryos developed normally during our imaging period and up to 48 hpf (Supplementary Fig. 4 and 5).".

Supplementary Figure 5. Development for HT-rad21 and HT-ctcf injected zebrafish embryos. Lateral views of whole embryos either uninjected or injected with HT-rad21 mRNA or HT-ctcf mRNA (see Methods). Stages are provided according to Kimmel et al., 1995. Scale bar is 500 μ m and applies to all images.

Do the tagged proteins compete with endogenous proteins for chromatin binding? ChIP-seq experiments in shield stage comparing anti-rad21/CTCF with anti-HT antibodies in injected embryos should provide some insight into this.

As suggested, we now controlled for chromatin binding of HT-tagged proteins. Reviewer 1 had a similar concern (point 1). We chose to perform subcellular protein fractionation experiments, which provide the required readout using protocols suited for early embryonic stages with limited cell numbers. Proteins able to bind chromatin are expected to show a band in the chromatin-bound fraction.

To confirm that HT-Rad21 or HT-CTCF indeed bind chromatin, we conducted subcellular protein fractionation experiments in shield-stage zebrafish embryos (see Supplementary Fig. 3 below). Since the low injection amount of 0.67 μ g of mRNA encoding for HT-Rad21 and 1.0 μ g of mRNA encoding for HT-CTCF in our single-molecule measurements did not allow for visualization of respective protein bands in Western blots, we injected 10-fold higher amounts than used for single-molecule measurements for these control experiments. All extracts (cytoplasmic (1-CE), membrane (2-ME), soluble nuclear (3-SNE), chromatin-bound (4-CBE), and cytoskeletal (5-CSE)) were probed using an anti-Rad21 or anti-CTCF antibody. We validated the chromatin-bound origin of the CBE by the presence of Histone H2B. In addition to the protein band corresponding to endogenous Rad21, we observed a higher-running band corresponding to HT-Rad21 (Supplementary Fig. 3 and Western blots quantifying HT-Rad21 overexpression, Supplementary Fig. 1, for band identification). Of note, the relative amounts of HT-Rad21 and HT-CTCF in each fraction closely recapitulate the distributions of endogenous Rad21 and CTCF. From these assays, we conclude that HT-Rad21 and HT-CTCF are indeed able to associate with chromatin to a similar extent as the endogenous proteins.

Supplementary Figure 3. Subcellular protein fractionation of HT-tagged wild-type proteins from lysates of shield-stage zebrafish embryos. a)-b) Western blots of subcellular protein fractionated shield-stage embryos injected in the 1-cell stage with a) 6.7 pg HT-rad21 mRNA or b) 10 pg HT-ctcf mRNA. Blots were probed with anti-Rad21 and anti-Histone H2B antibody (a), or anti-CTCF and anti-Histone H2B antibody (b). Embryos were injected with a 10-fold injection amount compared to our single-molecule measurements to enhance band clarity (see Methods). Lanes include extracts of: 1-CE: Cytoplasmic, 2-ME: Membrane, 3-SNE: Soluble nuclear, 4-CBE: Chromatin bound, 5-CSE: Cytoskeletal extracts.

We repeated the protein fractionation assay for HT-Rad21-3x (Supplementary Fig. 12a). This revealed a strong reduction of the chromatin-bound extract compared to HT-Rad21-WT relative to endogenous Rad21. This indicates that chromatin binding of HT-Rad21-3x is indeed impaired.

Similarly, the subcellular protein fractionation assay revealed chromatin binding of HT-CTCF and similar fraction distribution than endogenous CTCF (Supplementary Fig. 12b, c; for band identification, see Western blot Supplementary Fig. 1). In contrast, we could not detect the binding-deficient mutant HT-CTCF- Δ ZF4-7 in the chromatin-bound fraction.

Supplementary Figure 12. Subcellular protein fractionation of HT-tagged mutant proteins from lysates of shield-stage zebrafish embryos. a)-b) Western blots of subcellular protein fractionated shield-stage embryos injected in the 1-cell stage with a) 6.7 pg HT-rad21 3x mutant mRNA or b) 10 pg HT-ctcf Δ ZF4-7 mutant mRNA. Blots were probed with anti-Rad21 and anti-Histone H2B antibody (a), or anti-CTCF and anti-Histone H2B antibody (b). c) Intensity plot for Western blot bands depicted in panel b) between ~180-130 kDa (refer to grey marker lane), quantified across a straight region of each band (see Methods). Asterisk denotes the intensity maximum corresponding to HT-CTCF- Δ ZF4-7 protein. Embryos were injected with a 10-fold injection amount compared to our single-molecule measurements to enhance band clarity (see Methods). Lanes

include extracts of: 1-CE: Cytoplasmic, 2-ME: Membrane, 3-SNE: Soluble nuclear, 4-CBE: Chromatin bound, 5-CSE: Cytoskeletal extracts.

We added the results of the protein fractionation assays in the main text: "We further confirmed chromatin association of HT-Rad21 and HT-CTCF by subcellular protein fractionation (Supplementary Fig. 3)." and "We confirmed impaired chromatin binding of HT-Rad21-3x and HT-CTCF-ΔZF4-7 by subcellular protein fractionation (Supplementary Fig. 12)". We further added Supplementary Fig. 3 and 12, and a methodological description in the methods section.

Additionally, we now performed co-immunoprecipitation of Smc3 using an antibody against the HT-Rad21 fusion protein. Similar to what we described for reviewer 3 (point 2), we utilized the two 5' located HA-tags in our HT-Rad21 construct and performed a pulldown on shield-stage protein lysates from zebrafish embryos injected with HT-rad21 (we again used 6.7pg mRNA, 10-fold more compared to our single-molecule measurements, to allow for Western blot analysis). We validated the IP with a positive HA-tagged protein (Pierce™ HA-Tag IP/Co-IP Kit, Thermo Fisher Scientific) and uninjected zebrafish embryo lysate without HA-tag as a negative control. To unambiguously identify each protein, we ran equally distributed samples of a single IP on two blots (see below, new Supplementary Fig. 2a,b). In the first blot (Panel a), we verified the presence of the HA-tag in the input and pulldown of HT-rad21-injected, but not the uninjected WT control embryos (for band identification, see Western Blot Supplementary Fig. 1). Using an anti-Smc3 antibody, we identified the presence of Smc3 on the second blot in the Input samples of both HT-rad21-injected embryos and uninjected controls (Panel b). Importantly, Smc3 was only detected in the HA-tag pulldown of the HT-rad21-injected sample but not in the HA-tag pulldown of the uninjected control embryos (Panel b). To visualize both HT-Rad21 and Smc3 proteins on a single blot, we probed the second blot once more with the anti-HA tag antibody (Supplementary Fig. 1c). This indicates that HT-Rad21 is successfully incorporated into the cohesin complex.

Supplementary Figure 2: Co-immunoprecipitation (Co-IP) of Smc3 with HA-HT-Rad21 in zebrafish embryos. SDS-page gel stained with **a)** anti-HA tag antibody (Blot 1) and **b)** anti-Smc3 antibody (Blot 2). **c)** Blot 2 from panel b) subsequently stained with anti-HA tag antibody. Gels were loaded with: lysate of embryos injected with mRNA encoding for HA-tagged HT-Rad21 (lane 1, Input), immunoprecipitation (IP) using an anti-HA tag antibody (lane 2, IP HA), lysate of uninjected embryos (lane 3, WT, Input), IP using an anti-HA tag antibody (lane 4, IP HA). Endogenous Smc3 was detected in both the HA-HT-rad21 injected and uninjected WT input samples. Smc3 was also detected in the IP HA samples from injected embryos. HA-HT-Rad21 protein was only detected in embryos injected with the HA-HT-rad21 mRNA and not in the WT control.

We included our observation in the main manuscript: "We confirmed the incorporation of HT-Rad21 into the cohesin complex through co-immunoprecipitation of the N-terminally interacting Smc3 protein¹ (Supplementary Fig. 2)". We further added Supplementary Figure 2 and described the Co-IP in the methods section.

2- The overexpression of HT-rad21 and HT-CTCF is estimated to cause an increase of 10% and 25% in rad21 and CTCF protein expression, respectively. However, there is no quantification in the western blots of Suppl. Fig. 1a-b, and its figure legend states that the injection amounts were increased 10-fold compared to the single-molecule measurements to enhance band clarity. Therefore, how did the authors calculate such 10% and 25% increase in protein expression? Also, CTCF protein is maternally provided in zebrafish embryos (see PMID 34518536); is rad21 also maternally provided?

We now extended Western blots to three biological replicates and provide quantification graphs alongside our Western blots (see below, Supplementary Fig. 1, see also comment 1 of reviewer 3). Since the low injection amounts of mRNA we used in single-molecule experiments did not allow for visualization of the fusion protein bands in Western blots, we injected 10-fold higher amounts than used for single-molecule experiments to determine the expression level, then divided the value by 10. For clarity, we now adapted the quantification plots to reflect the raw Western blots. From the Western blots we inferred an overexpression of $(114 \pm 26)\%$ for HT-Rad21 and $(240 \pm 27)\%$ for HT-CTCF (Supplementary Fig. 1b,d). Considering that we used 10-fold smaller injection amounts in our single-molecule measurements, we obtain an overexpression of $(11 \pm 3)\%$ for HT-Rad21 and $(24 \pm 3)\%$ for HT-CTCF in our single-molecule measurements.

Supplementary Figure 1. Quantification of HT-Rad21 and HT-CTCF expression levels in shield-stage zebrafish embryos. a), c) Western blots (WB) of three independent biological replicates of uninjected embryos (WT) or embryos injected with 10-fold increased injection amounts compared to our single-molecule measurements to enhance band clarity (see Methods). **a)** WB with anti-Rad21 antibody of embryos injected with 6.7 pg mRNA encoding for HT-rad21 (HT-rad21) or uninjected embryos (WT). **b)** Quantification of a) comparing the HT-Rad21 band to the WT-Rad21 band. Single-molecule injections were performed with 10-fold decreased mRNA amount yielding $(11 \pm 3)\%$ of ectopic HT-Rad21 expression. **c)** WB with anti-CTCF antibody of embryos injected 10 pg mRNA encoding for HT-ctcf (HT-ctcf) or uninjected embryos (WT). **d)** Quantification of c) comparing the HT-CTCF band to the WT-CTCF band. Single-molecule injections were performed with 10-fold decreased mRNA amount yielding $(24 \pm 3)\%$ of ectopic HT-CTCF expression. Lines represent mean values \pm s.d.

We added our calculation procedure to the methods section: “Quantification was performed with Image Lab 6.0 by dividing the background subtracted HT-protein band intensity with the endogenous protein band intensity. Since we injected 10-fold less mRNA for single-molecule measurements, we divided the value by a factor of 10 to obtain the expression levels in single-molecule experiments.”

Indeed, maternal supply of Rad21 is reported in the literature^{20,21}. We now mention maternal supply of Rad21 and CTCF in the manuscript: “..., which resulted in ectopic protein expression of ca. $(11.4 \pm 2.6)\%$ for HT-Rad21 and $(24.0 \pm 2.7)\%$ for HT-CTCF of endogenous and maternally^{10,20,21} provided protein in shield-stage embryos (Supplementary Fig. 1)”

3- The mutant rad21 and CTCF proteins used here reproduce previously reported mutations, but have you tested their effect on DNA binding in zebrafish embryos?

As suggested, we controlled for chromatin binding of HT-Rad21-3x and HT-CTCF-ΔZF4-7 using a Subcellular protein fractionation assay (see response to comment 2 and Supplementary Fig. 12). In summary, we observed strong decrease in chromatin association for HT-rad21-3x compared to HT-Rad21 and could not detect HT-CTCF-ΔZF4-7 in the chromatin-bound fraction.

4- In Fig. 1i, it is intriguing that inhibition of RNAPII elongation reduces both the long and short-bound fractions of rad21; if the long-bound fraction is interpreted as DNA bound molecules and the short-bound fraction as unspecific transient interactions, how do you explain that short-bound molecules also increase upon transcriptional elongation inhibition?

We do not know the molecular origin of HT-Rad21 short-bound molecules. Short binding events persist for chromatin-binding-impaired HT-Rad21-3x (Fig. 1h and new control experiments, Supplementary Fig. 12) and upon modulation of Nipbl or CTCF expression levels (Fig. 1i). Thus, short HT-Rad21 binding events might represent transient unspecific interactions with DNA, RNA, proteins or other biomolecules or spatial confinement. Such interactions could still be prone to elongation-based disturbance.

We now more clearly state our limited understanding of the short-bound Rad21 fraction in the manuscript: "...while short-bound fractions are presumably associated with unspecific transient modes of binding to DNA, RNA, proteins or other biomolecules, or transient spatial confinement." And discussion: "The origin of short binding events of CTCF and Rad21 were less clear but presumably represented transient unspecific interactions to DNA, RNA, proteins or other biomolecules, or transient spatial confinement."

5- Regarding the polymer models simulating the effect of cohesin and CTCF on chromatin architecture, the results of the simulation show an establishment of this structure higher than the real HiC data. For instance, Fig. 4b shows a higher increase in contact probability at shorter distances during development than the increase observed in data from Wike et al (green arrowheads). Moreover, this effect is more notable in Fig. 4g, where one can see that insulation around CTCF sites in shield stage is much more prominent for the simulation than in the real HiC. How do you explain these differences? Are there any other factors that counteract the formation of chromatin structure in vivo? Are you considering all CTCF sites for calculating insulation or only those falling within TAD boundaries (if they can be calculated in shield stage)?

The reviewer is right, the increase in contact probabilities at short distances is more pronounced in simulations compared to experiments. We would like to emphasize however that the aim of our simulations was not to perfectly reproduce the experimental Hi-C maps across all developmental stages but rather to provide insight whether the experimentally observed changes in binding kinetics of cohesin and CTCF, could in principle, support a trend toward more structured chromosomes as observed in the Hi-C experiments.

For this reason, in the 1-kb simulations of cohesin action (Figure 4a-f) we did not include CTCF and the loop-extrusion blocking activities of other complexes, such as PolII^{22,23} and the MCM complex²⁴, nor any other features that can pause or compete with loop extrusion activity. In addition, the polymer system we used as a basis for our loop extrusion simulations is a low-density coil in a theta solvent, which has a steeper scaling of contact probability (~-1.5) than what was observed experimentally for

(mammalian) chromosomes after depletion of loop extrusion factors (~ -1)^{25,26}. We attribute the more pronounced increase in contact probability to these factors.

In 10kb simulations, we considered both cohesin and CTCF molecules. In our analysis we considered all CTCF sites within open chromatin regions that match the motif. Since the permeability of these sites to loop extrusion is a priori variable and unknown, we did not account for such variability and simulated a simplified model with impermeable sites. Again, Figure 4g shows structural features appearing in the simulations at short distances, which are less visible in the real Hi-C data.

We now better explain the aim of our simulations in the manuscript: “To obtain insight into whether the experimentally observed changes in binding kinetics of cohesin and CTCF could support a trend toward more structured chromosomes as observed in the Hi-C experiments, we turned to polymer simulations of cohesin-mediated loop extrusion. “

We further extended the discussion of our simulations and the differences to the Hi-C experiments in the discussion section: “We attribute the more pronounced increase in contact probability of our simulations to omitting factors that can pause or compete with loop extrusion activity and to using a low-density coil in a theta solvent, which has a steeper scaling of contact probability (~ -1.5) than what was observed experimentally for mammalian chromosomes after depletion of loop extrusion factors (~ -1)^{25,26}.”

b) Minor points:

1- Please, increase the size of Figure 1i.

As suggested, we increased the size.

2- HT-CTCF data is missing from Suppl. Fig. 5a for comparison with the HT control.

We now added an additional panel to the Figure, allowing comparison of HT-CTCF with the HT control (see Supplementary Fig. 10b below).

Supplementary Figure 10b: HaloTag (HT)-control for HT-CTCF recorded with interlaced time-lapse microscopy (ITM). Fractions of short and long binding events of HT control or HT-CTCF. Data represent mean \pm s.d. of movie-wise determined fractions. Insets show zooms into the respective graphs. Lines serve as guides to the eye.

3- In Fig. 2b, short and long bound HT-rad21 molecules should be in the same graph and the statistics for their comparison indicated as referred in the text.

We thank the reviewer for the suggestion and have revised Fig. 2b (shown below) to include short and long bound molecules including additional statistics in a single graph. Additional statistics are provided in Supplementary Tables 10-11.

Figure 2b: Mean jump distances of HT-Rad21 molecules classified as short bound (circles) and long bound (squares) at different developmental phases. pre-ZGA (red): 64-, 128-, 256, 512-cell stages pooled; post-ZGA (green): high, oblong, sphere stages pooled; shield stage (blue) and shield stage with the addition of nipbl-morpholino (MO, cyan). Insets: example tracks. Data represent mean \pm s.d. P-values were calculated using the Kruskal-Wallis-Test and Multiple Mann-Whitney-Test. Scale bar: 1 μ m.

4- Please, reorder the labeling of panels in Fig 4 according to the order in which they are cited in the text.

We adjusted the labeling order of Fig. 4 as suggested.

Thank you.

Reviewer #3 (Remarks to the Author):

This paper by Cossmann et al. addresses an interesting question regarding the dynamics of CTCF and cohesin association with chromatin during zebrafish development. CTCF and cohesin play key roles in organizing genomes in 3D as well as in repair and sister chromatid cohesion and are of high current interest. To my knowledge, how the chromatin association dynamics of these proteins change during embryo development has not been studied before this paper, making the contribution novel. Moreover, the methodological approaches are quite cutting-edge combined single-molecule imaging in live zebrafish embryos, what appears to be thoughtful analysis of the single-molecule data and also simulations.

Therefore, I think this paper could be of general interest to the wide readership of Nature Communications and a nice contribution to the literature. However, I do have one quite major concern about the extremely low DNA-bound fraction of cohesin and the lack of consideration of sister-chromatid cohesion, as well as several other comments and concerns, I would like to see the authors address. If the authors can fully address these points, I think it could be a nice paper and contribution to the literature.

We thank the reviewer for the positive feedback and constructive criticism on our manuscript. We thoroughly addressed all concerns in our point-by-point responses and the revised manuscript.

MAJOR CONCERN: Sister chromatid-cohesion and cohesive cohesin Cohesin has several functions, including 1) sister-chromatid cohesion and 2) loop extrusion (and others like repair etc.). In this paper, the authors exclusively interpret their results in terms of loop extrusion, but cohesion is required for nuclear division and arguably the most important function of cohesin. I am concerned about the extremely low bound fraction of HT-rad21. If I am reading Supp Table S2 correctly, the long bound fraction at 128-cell stage is 0.0002. Zebrafish has 25 chromosomes, so 50 diploid chromosomes. Let's assume we need at least 100 cohesive cohesins per chromosome for proper chromosome segregation at nuclear division, this means we need at least 5000 long-bound cohesins (cohesive cohesin is known to be "infinitely stable", Gerlich 2006). If the long-bound fraction is 0.0002, this means there needs to be $5000/0.0002 = 25$ million cohesin complexes per cell at least which is completely unreasonable.

Even if we assume the long-bound fraction is only 0.003, we still need 1.67 million cohesins per nucleus. Can the authors please estimate the per-cell abundance of cohesin?

I think a plausible explanation for how their observations fit with the known essential cohesion function of cohesin is required.

We thank the reviewer for their thoughtful analysis. However, the extremely high amount of 25 million cohesin complexes is based on a misunderstanding of the values given in Suppl. Table 2. These values represent statistical P-values for inter-stage differences in the data shown in Fig. 1h, not the actual bound fractions. We apologize for the misleading table caption and changed the captions of affected Supplementary Tables for clarity. The values of bound fractions are published in form of source data in the source data file.

Following the approach of the reviewer, we reestimated the total number of cohesins per nucleus. Using the lowest long-bound HT-Rad21 fraction of 0.44% (256-cell stage) and assuming 5000 long-bound cohesive cohesins required for chromosome segregation yields approximately 1.1 million cohesins per nucleus. This value is still larger than a previous estimate of 508,000 nuclear Scc1 cohesins in Prometa phase-synchronized HeLa cells²⁷ and an estimate of approximately 109,000 Rad21

molecules in mESC cells²⁸. However, Rad21 is maternally provided to compensate for transcriptional quiescence in the embryo before ZGA, which starts after ca. 10 cell divisions^{20,21}. Maternally provided proteins typically exceed the need of a single cell, as they are diluted by successive cell divisions. This process might well account for the very low bound fraction observed in early developmental stages.

We included this explanation of the low bound fraction in early stages in the discussion section of the manuscript.

Also, what is the time-scale of nuclear division throughout the zebrafish developmental stages studied here? I am also concerned about the 100-second reported residence time for cohesin, since cohesive cohesin is known to be extremely stable (Gerlich 2006) as opposed to loop extruding cohesin which has residence times of 15-30 min in mammalian systems.

We agree with the reviewer that the long-bound residence time of 100s is surprisingly short compared to the observation of much longer times in mammalian systems. Indeed, it is also short compared to nuclear division times, which are 15-25 min for pre-ZGA stages (64-, 128-, 256-, 512-cell) consisting of S and M phases and 30-60 min for post-ZGA stages (high, oblong, sphere), which include G2 phases. In shield stage, nuclear divisions are on the order of hours with more prominent G1 phases^{29,30}.

In our single-molecule data we observe individual bound molecules for up to 4 min. Due to experimental limitations such as photobleaching and movement of nuclei in the living embryo, we cannot exclude that a fraction of molecules stays bound for longer times. We now more clearly state that the longest residence time cluster of 100 s we obtain by GRID analysis of survival time distributions in shield-stage embryos represents a lower limit. However, we note that we measure even shorter residence times for pre-ZGA (52 ± 6 s) and post-ZGA (63 ± 2 s) stages.

Beyond making the numbers match, I'd also like to ask the authors to go through the text and interpret their cohesin results both through the lens of cohesions and extrusion, instead of exclusively focusing on extrusion.

We agree that our focus was too exclusive. Indeed, it might be suspected that most long bound cohesin molecules pre-ZGA are involved in cohesive activity, given the absence of G1 phases in early zebrafish. However, the need for cohesive cohesins is not expected to change during development. Thus, even if all long-bound HT-Rad21 molecules we observed in early developmental stages were involved in cohesion, the increase in the bound fraction during development suggests at least partial involvement of Rad21 in functions of cohesin other than cohesion.

As suggested, we now also discuss our results with regard to cohesion.

MAJOR COMMENTS

1. PROTEIN LEVELS: line 65 says HT-rad21 and HT-CTCF increases the total rad21 and TCF abundance by 10% and 25% respectively. But when I look at the Westerns in Fig. S1a,b it looks like the HT-rad21 band is about same as wt-rad21 (so 100% increase) and when I look at HT-CTCF it looks like at least 5x (so 500% increase). Can the authors please include careful quantification (including replicate-to-

replicate error bars (similar to Fig. S1d)). Getting these numbers right is crucial for interpretations of the rest of the paper.

We understand the concern of the reviewer. Reviewer 2 had a similar remark (point 2). Importantly, the amount of mRNA we injected for the single-molecule experiments did not allow for quantification of expression levels via Western blots, since the signal of the ectopic HT-Rad21 or HT-CTCF band was too low. For quantification of expression levels, we therefore injected 10-fold more mRNA. This explains the discrepancy between the expression levels seen in Western blots and those we report for single-molecule experiments.

As suggested, we now performed in total three biological replicates to quantify HT-protein expression levels via Western blot of 10x-injected embryos (see Supplementary Fig. 1 below). For clarity, we now adapted the quantification plots to reflect the raw Western blots. From the 10-fold overinjected lysates we inferred an overexpression of $(114 \pm 26)\%$ for HT-Rad21 and $(240 \pm 27)\%$ for HT-CTCF. Considering that we used 10-fold smaller injection amounts in the actual experiment, we can infer an overexpression of $(11 \pm 3)\%$ for HT-Rad21 and $(24 \pm 3)\%$ for HT-CTCF in our single-molecule measurements.

We added the Western blots and corresponding quantification as Supplementary Fig. 1 and cite the obtained values in the main manuscript.

Supplementary Figure 1. Quantification of HT-Rad21 and HT-CTCF expression levels in shield-stage zebrafish embryos. a), c) Western blots (WB) of three independent biological replicates of uninjected embryos (WT) or embryos injected with 10-fold

increased injection amounts compared to our single-molecule measurements to enhance band clarity (see Methods). **a)** WB with anti-Rad21 antibody of embryos injected with 6.7 pg HT-rad21 mRNA or uninjected embryos (WT). **b)** Quantification of a) comparing the HT-Rad21 band to the endogenous Rad21 band. Single-molecule injections were performed with 10-fold decreased mRNA amount yielding $(11 \pm 3)\%$ of ectopic HT-Rad21 expression. **c)** WB with anti-CTCF antibody of embryos injected 10 pg HT-ctcf mRNA or uninjected embryos (WT). **d)** Quantification of c) comparing the HT-CTCF band to the endogenous CTCF band. Single-molecule injections were performed with 10-fold decreased mRNA amount yielding $(24 \pm 3)\%$ of ectopic HT-CTCF expression. Lines represent mean values \pm s.d.

2. IS RAD21 FUNCTIONAL? Along the lines of #1, when you over-express a sub-unit of a multi-protein complex, the over-expressed sub-unit may not be functional. It seems the authors did N-terminal HT tag on rad21 which is unusual (most paper do C-term). How did the authors validate the HT-rad21 is functional and incorporated in cohesin complexes along with the other cohesin subunits?

We thank the reviewer for addressing this critical point, which was also raised by Reviewer 1 (point 1). As suggested, we now performed several control experiments.

To confirm that HT-Rad21 and HT-CTCF indeed bind chromatin, we conducted subcellular protein fractionation experiments in shield-stage zebrafish embryos injected in the 1-cell stage. We again injected 10-fold higher HT-rad21 mRNA amounts (6.7 pg) or HT-CTCF mRNA amounts (10 pg) compared to our single molecule experiments (0.67 pg and 1.0 pg, respectively), to allow for Western blot analysis. All extracts (cytoplasmic (1-CE), membrane (2-ME), soluble nuclear (3-SNE) and cytoskeletal (5-CSE)) were probed using an anti-Rad21 or anti-CTCF antibody. We validated the chromatin bound origin of the chromatin-bound extract (4-CBE) by the presence of Histone H2B. In addition to the expected endogenous protein bands we observed HT-Rad21 and HT-CTCF bands in all extracts (see Supplementary Fig.3a below, and Western blots quantifying HT-Rad21 overexpression (Supplementary Fig. 1) for band identification). Of note, the relative amounts of HT-Rad21 and HT-CTCF in each fraction closely recapitulate the distributions of endogenous Rad21 and CTCF. From these assays, we conclude that HT-Rad21 and HT-CTCF are indeed able to associate with chromatin to a similar extent as the endogenous proteins.

Supplementary Figure 3. Subcellular protein fractionation of HT-tagged wild-type proteins from lysates of shield-stage zebrafish embryos. a)-b) Western blots of subcellular protein fractionated shield-stage embryos injected in the 1-cell stage with **a)** 6.7 pg HT-rad21 mRNA or **b)** 10 pg HT-ctcf mRNA. Blots were probed with anti-Rad21 and anti-Histone H2B antibody (a), or anti-CTCF and anti-Histone H2B antibody (b)). Embryos were injected with a 10-fold injection amount compared to our single-molecule measurements to enhance band clarity (see Methods). Lanes include extracts of: 1-CE: Cytoplasmic, 2-ME: Membrane, 3-SNE: Soluble nuclear, 4-CBE: Chromatin bound, 5-CSE: Cytoskeletal extracts.

We repeated the protein fractionation assay for HT-Rad21-3x (Supplementary Fig. 12a). This revealed a strong reduction of the chromatin-bound fraction of the HT-Rad21-3x mutant compared to HT-Rad21-WT, normalized to endogenous Rad21. Thus, we conclude that chromatin binding of HT-Rad21-3x is indeed impaired, as reported^{2,3,5}. Similarly, the binding-deficient mutant HT-CTCF- Δ ZF4-7 did not show chromatin binding (Supplementary Fig. 12b, c; for band identification, see Western blot Supplementary Fig. 1).

Supplementary Figure 12. Subcellular protein fractionation of HT-tagged mutant proteins from lysates of shield-stage zebrafish embryos. a)-b) Western blots of subcellular protein fractionated shield-stage embryos injected in the 1-cell stage with a) 6.7 pg HT-rad21-3x mutant mRNA or b) 10 pg HT-ctcf- Δ ZF4-7 mutant mRNA. Blots were probed with anti-Rad21 and anti-Histone H2B antibody (a)), or anti-CTCF and anti-Histone H2B antibody (b)). c) Intensity plot for Western blot bands depicted in panel b) between ~180-130 kDa (refer to grey marker lane), quantified across a straight region of each band (see Methods). Asterisk denotes the intensity maximum corresponding to HT-CTCF Δ ZF4-7 protein. Embryos were injected with a 10-fold injection amount compared to our single-molecule measurements to enhance band clarity (see Methods). Lanes include extracts of: 1-CE: Cytoplasmic, 2-ME: Membrane, 3-SNE: Soluble nuclear, 4-CBE: Chromatin bound, 5-CSE: Cytoskeletal extracts.

Overall, the results from the chromatin fractionation assays confirmed chromatin binding of HT-Rad21 and HT-CTCF and impaired chromatin binding of HT-Rad21-3x and HT-CTCF- Δ ZF4-7. Together with our observation that impaired chromatin binding affects the long-bound fraction but not the short-bound fraction of HT-Rad21 and HT-CTCF, respectively, this indicates that long-binding events represent a chromatin binding mode of wild-type proteins, while short-bound proteins represent other modes of binding.

We further performed co-immunoprecipitation of Smc3 using an antibody against the HT-Rad21 fusion protein. We utilized the two 5' located HA-tags in our HT-Rad21 construct and performed a pulldown on shield-stage protein lysates from zebrafish embryos injected with HT-rad21 (we again used 6.7pg mRNA, 10-fold more compared to our single-molecule measurements, to allow for Western blot analysis). We validated the IP with a positive HA-tagged protein (Pierce™ HA-Tag IP/Co-IP Kit, Thermo Fisher Scientific) and uninjected zebrafish embryo lysate without HA-tag as a negative control.

Supplementary Figure 2: Co-immunoprecipitation (Co-IP) of Smc3 with HA-HT-Rad21 in zebrafish embryos. SDS-page gel stained with **a)** anti-HA tag antibody (Blot 1) and **b)** anti-Smc3 antibody (Blot 2). **c)** Blot 2 from panel b) subsequently stained with anti-HA tag antibody. Gels were loaded with: lysate of embryos injected with mRNA encoding for HA-tagged HT-Rad21 (lane 1, Input), immunoprecipitation (IP) using an anti-HA tag antibody (lane 2, IP HA), lysate of uninjected embryos (lane 3, WT, Input), IP using an anti-HA tag antibody (lane 4, IP HA). Endogenous Smc3 was detected in both the HA-HT-rad21 injected and uninjected WT input samples. Smc3 was also detected in the IP HA samples from injected embryos. HA-HT-Rad21 protein was only detected in embryos injected with the HA-HT-rad21 mRNA and not in the WT control.

To unambiguously identify each protein, we ran equally distributed samples of a single IP on two blots (see below, new Supplementary Fig. 2a,b). In the first blot (Panel a), we verified the presence of the HA-tag in the input and pulldown of HT-rad21-injected, but not the uninjected WT control embryos (for band identification, see Western Blot Supplementary Fig. 1). Using an anti-Smc3 antibody, we identified the presence of Smc3 on the second blot in the Input samples of both HT-rad21-injected embryos and uninjected controls (Panel b). Importantly, Smc3 was only detected in the HA-tag pulldown of the HT-rad21-injected sample but not in the HA-tag pulldown of the uninjected control embryos (Panel b). To visualize both HT-Rad21 and Smc3 proteins on a single blot, we probed the second blot once more with the anti-HA tag antibody (Supplementary Fig. 2c). This indicates that HT-Rad21 is successfully incorporated into the cohesin complex.

We further measured the bound fractions of a C-terminally tagged Rad21 fusion, Rad21-HT, in single-molecule experiments (see Supplementary Fig. 14 below). Rad21-HT showed bound fractions comparable to the N-terminally tagged HT-Rad21 at most developmental stages and a similar trend for both the long- and short-bound fractions. This indicates that our choice of an N-terminal tag is not the origin of the small bound fractions at early developmental stages. Of note, while indeed many studies utilized a C-terminal tag for Rad21, they did not provide an explicit rationale for this choice, although both N- and C-termini of Rad21 are involved in Rad21s function within the cohesin complex^{1,3,31-33}. Moreover, it was reported that a C-terminal tag might interfere with two-ring cohesin formation³¹.

Supplementary Figure 14. Comparison of bound fractions of HT-Rad21 and Rad21-HT. Fractions of short and long binding events of Rad21-HT or HT-Rad21 obtained by ITM measurements. Data represent mean \pm s.d. of movie-wise determined fractions. Inset shows zoom into the respective graph. Lines serve as guides to the eye.

We added the ColP, subcellular protein fractionation, and bound fraction measurements for Rad21-HT as Supplementary Fig. 2, 3, 12, and 14 to the manuscript. In the methods section, we further described the ColP and subcellular protein fractionation assay for zebrafish embryos.

3. PROVIDE DATA? I appreciate the authors making the raw data available, but when I clicked on https://datadryad.org/stash/share/zf4_oJgrWv-741Xb_oSMaNTmYbHZhe5FDHdPz6QRumltl I get a “403 Forbidden” error message. Maybe the provided URL is incorrect, or maybe I am doing something incorrect. Could you provide an working URL to the data?

We thank the reviewer for taking the time to validate our raw data. There seems to be a copy-paste error, as the link the reviewer used contains the line number. We kindly recommend manually copying the URL:

https://datadryad.org/stash/share/zf4_oJgrWv-Xb_oSMaNTmYbHZhe5FDHdPz6QRumltl

4. SECTION LINE 110-129 appears to be missing figure references? Please insert references to the figure panels? For Pol II inhibition, what was the treatment length? Presumably Pol II inhibition is lethal? How long can Pol II be inhibited before the animal dies or serious artifacts occur?

We thank the reviewer for pointing this out. We have now added the appropriate references.

We performed PolII inhibition according to published protocols: α -Amanitin was co-injected in the 1-cell stage, and Triptolide-treated embryos were immersed in Triptolide solution following dye incubation around the 4- to 8-cell stages. Consequently, the treatment length before shield stage measurements was approximately 9 hours for α -Amanitin and 8 hours for Triptolide. We have added this information in the methods section.

As suggested, we now performed morphological analysis of embryos treated with α -Amanitin and Triptolide (see Supplementary Fig. 21 below). Both treatments resulted in developmental arrest after zygotic genome activation around the 50%-epiboly stage. This is comparable to previous publications for α -Amanitin^{34–37} and Triptolide^{38–40} treatment. Furthermore, embryos of both treatments did not reach the 24 hours post-fertilization stage. We have clarified this observation in the main text.

Supplementary Figure 21: Development of α -Amanitin and Triptolide treated zebrafish embryos. Lateral views of whole embryos either uninjected (WT) or treated with α -Amanitin or Triptolide (see Methods). Embryos injected with α -Amanitin or Triptolide exhibited developmental delays, remaining at around sphere stage, compared to the time point of shield stage in uninjected embryos. Both injections were lethal at 24 hours post-fertilization (hpf). Stages are provided according to Kimmel et al., 1995. Scale bar is 500 μ m and applies to all images.

We now mention our observations in the main manuscript and added Supplementary Fig. 21.

5. LINE 145: The NIPBL perturbation is interesting, but it is exclusively interpreted in terms of extrusion. Schwarzer...Spitz Nature 2017 showed that del-NIPBL removes all cohesin from chromatin, consistent with NIPBL loading cohesin. How much does the DNA-bound population of HT-rad21 change? Can you please compare to Schwarzer 2017?

The reviewer is right that the increased mobility of long-bound cohesin in presence of nipbl-MO might, in addition to decreased loop extrusion, also be due to an overall lower amount of cohesin bound to chromatin. A similar comment was given by Reviewer 1 (comment 5). We have so far overlooked this possibility since we characterized the mobility of local cohesin-bound loci. Indeed, multiple publications demonstrated a reduced association of cohesin to chromatin upon lower Nipbl levels by ChIP-seq, comparable to Schwarzer et al. 2017¹¹⁻¹⁵. We thank the reviewer for highlighting this additional mechanism of increasing chromatin mobility. We now highlight in the manuscript potential contributions from both reduced cohesive function and reduced Nipbl-mediated ATPase activity at lower Nipbl levels: "Since Nipbl both reduces chromatin-bound cohesin¹¹⁻¹⁵ and stimulates the ATPase activity of cohesin¹⁶⁻¹⁹, both a reduced cohesive function and reduced loop extrusion activity might contribute to increased chromatin mobility."

6. LINE 210-245 SIMULATIONS: The simulations are interesting, but reported in an unconventional manner (association rates). To make it easier to compare with other loop extrusion polymer simulation studies can the authors also report it in the standard way: extruder processivity, density/separation, speed (they do report speed). Can they please include the best fit of these numbers in main Fig 4 and in the main text? Can they also please compare to mammalian systems? The arrowheads in Fig. 4b,c,e,f would benefit from having the x-axis value listed (e.g. ~300kb).

As suggested, we improved on reporting our simulations.

We calculated the values for speed and processivity according to⁴¹ with the following equations: Speed (S) = 0.5 kb/sec and Processivity (P) = Speed * Residence time

Due to the lack of reference values in zebrafish, we based the value for cohesin speed used in the simulations on known mammalian parameters ranging from 0.1 kb/s²⁸ to 1 kb/s¹⁶. This yielded the following values:

Pre-ZGA stage:

Speed= 0.5 [kb/sec]

Residence time= 50 [sec]

Processivity = 25 [kb]

shield stage:

Speed= 0.5 [kb/sec]

Residence time= 100 [sec]

Processivity = 50 [kb]

We calculated the density using our residence times and relative changes in search time. This yielded a density of 0.7 Mb⁻¹ in pre-ZGA and of 33.3 Mb⁻¹ in shield stage. The values are comparable to other experimental estimates of 2.5-5.1 Mb⁻¹^{2,28}.

We now explicitly give the values for speed, processivity and density in the text and in Figure 4c.

As suggested, we now also added values of genomic distance to the arrowheads in Figure 4c-f.

7. In the Movies, which look really nice by the way, why are there so frequently molecules outside the nucleus?

As mentioned above, we now performed subcellular protein fractionation assays to control for chromatin association of HT-Rad21 and HT-CTCF. These experiments confirmed the presence of both the endogenous and the tagged proteins in the nucleoplasmic and the chromatin-bound fraction. In addition, the assays also showed a cytoplasmic fraction of these proteins, consistent with our single-molecule measurements of tagged proteins.

A cytoplasmic fraction of HT-Rad21 and HT-CTCF is not unexpected: cohesin is known to accumulate in the cytoplasm following chromosome dissociation at the end of mitosis⁴² and requires transport to reenter the nucleus, with a subset potentially remaining in the cytoplasm. Moreover, proteins are synthesized in the cytoplasm prior to nuclear transport, and we might observe such cytoplasmic proteins.

SMALLER MINOR COMMENTS

i. Line 41: along with Ref 6,25, authors should also cite de Wit Mol Cell 2015 for CTCF polarity.

We have added the citation to our manuscript.

ii. Really small thing, but why is rad21 all lower case but CTCF all upper case throughout the paper?

We adhered to the standard nomenclature of writing CTCF in capital letters, reflecting its acronymic nature as the CCCTCF-binding factor. While we inject HT-rad21 RNA, we actually measure HT-Rad21 protein. We thank the reviewer and have revised all occurrences throughout the manuscript to reflect the appropriate context-dependent usage.

iii. Fig 1d,e shows a lot of data, but really difficult to see by eye whether a fraction is 0.03 or 0.06 because it is so zoomed out. Can the authors start the y-axis at 0.01 or show a zoom-in inset or something along those lines so we can see the values?

We appreciate the reviewers concern regarding the data visualization in Fig 1d, e (shown below). We opted for a logarithmic scale to better represent the data across a wide range. As suggested, we now start the y-axis at 0.004 to include all error bars for HT-Rad21. To maintain consistency, we applied the same axis parameters to HT-CTCF.

Figure 1d, e: Fractions of a three-component diffusion model fitted to jump distance distributions of HT-Rad21 and HT-CTCF.

We kindly refer to the comprehensive Source Data file for precise analysis and independent visualization of the data.

iv. Line 76-78: Sorry if I missed it, but can you please provide the diffusion coefficients associated with each fraction?

The diffusion coefficients for each fraction and stage are represented in Supplementary Fig. 8, which we've included below for reference:

Supplementary Figure 8: Mobility of HT-Rad21 and HT-CTCF during zebrafish development. Diffusion coefficients of **a)** HT-Rad21 and **b)** HT-CTCF obtained from a three-component diffusion model fitted to jump distance distributions. Colors indicate stages of development (64-, 128-, 256-, 512-, 1k-cell, high, oblong, sphere, shield). Bars represent mean values \pm s.d. from 500 resamplings using 80% of randomly selected jump distances.

You also mention anomalous diffusion – how did you determine that the diffusion is anomalous, what kind (e.g. fraction Brownian motion?) and what alpha did you measure?

We apologize for not having been clear enough. We do not infer anomalous diffusion from our fit but assume anomalous diffusion of molecules in the nucleus based on previous reports^{43–46}. Moreover, bound molecules appear as slowly diffusing mainly due to limited localization precision in each frame and slow motion of the underlying chromatin. The apparent jump distances connecting successive error-afflicted localizations are Gaussian distributed and therefore resemble Brownian diffusion with a certain apparent diffusion coefficient. For simplicity and in accordance with previous work, we used a three-component Brownian diffusion model to describe the data^{47–50}. The first component fits the apparent Brownian motion of bound molecules and slow chromatin motion, while the remaining two components are used to approximate the anomalous diffusion of mobile molecules. We now better explain our analysis in the manuscript.

v. Line 112, Authors should probably cite Tedeschi Nature 2013 for WAPL

We have added the article to the text.

vi. Line 150 “increase in chromatin architecture” is a bit vague – at least I do not know what it means. Can you rephrase?

As suggested, we now detail in the main text what we mean by increased chromatin architecture: “The decrease in mobility of HT-Rad21-bound chromatin is compatible with loops and TADs becoming more prominent during embryo development.”

And discussion: “...and compatible with an increase in chromatin structures such as compartments, TADs, or loops during embryo development.”

vii. Line 291, Ref 88 is a bit messy and difficult to interpret since long-term partial Pol II depletion can have all sorts of indirect and secondary effects. Probably Banigan PNAS 2023 is a better reference.

We thank the reviewer for the suggestion and exchanged the references.

Reviewer #3 (Remarks on code availability):

The data URL does not work, so although I tried, I could not evaluate since the provided URL is not working.

As we have commented above, this appeared to be due to a copy-paste error, which included the line number. We kindly recommend manually copying the URL:

https://datadryad.org/stash/share/zf4_oJgrWv-Xb_oSMaNTmYbHZhe5FDHdPz6QRumItl

References

1. Haering, C. H., Löwe, J., Hochwagen, A. & Nasmyth, K. Molecular Architecture of SMC Proteins and the Yeast Cohesin Complex C-terminal domains forming a head would be part of. *Mol. Cell* **9**, 773–788 (2002).
2. Hansen, A. S., Pustova, I., Cattoglio, C., Tjian, R. & Darzacq, X. CTCF and cohesin regulate chromatin loop stability with distinct dynamics. *Elife* **6**, (2017).
3. Haering, C. H. *et al.* Structure and stability of cohesin's Smc1-kleisin interaction. *Mol. Cell* **15**, 951–964 (2004).
4. Davidson, I. F. *et al.* Rapid movement and transcriptional re-localization of human cohesin on DNA. *EMBO J.* **35**, 2671–2685 (2016).
5. Sun, Y. *et al.* RAD21 is the core subunit of the cohesin complex involved in directing genome organization. *Genome Biol.* **24**, 155 (2023).
6. Stainier, D. Y. R. *et al.* Guidelines for morpholino use in zebrafish. *PLoS Genet.* **13**, (2017).
7. Muto, A., Calof, A. L., Lander, A. D. & Schilling, T. F. Multifactorial origins of heart and gut defects in Nipbl-deficient zebrafish, a model of cornelia de Lange Syndrome. *PLoS Biol.* **9**, (2011).
8. Delgado-Olguín, P. *et al.* CTCF promotes muscle differentiation by modulating the activity of myogenic regulatory factors. *J. Biol. Chem.* **286**, 12483–12494 (2011).
9. Meier, M. *et al.* Cohesin facilitates zygotic genome activation in zebrafish. *Dev.* **145**, 1–13 (2018).
10. Franke, M. *et al.* CTCF knockout in zebrafish induces alterations in regulatory landscapes and developmental gene expression. *Nat. Commun.* **12**, (2021).
11. Schwarzer, W. *et al.* Two independent modes of chromatin organization revealed by cohesin removal. *Nature* **551**, 51–56 (2017).
12. Alonso-Gil, D., Cuadrado, A., Giménez-Llorente, D., Rodríguez-Corsino, M. & Losada, A. Different NIPBL requirements of cohesin-STAG1 and cohesin-STAG2. *Nat. Commun.* **14**, 1–11 (2023).
13. Newkirk, D. A. *et al.* The effect of Nipped-B-like (Nipbl) haploinsufficiency on genome-wide cohesin binding and target gene expression: Modeling Cornelia de Lange syndrome. *Clin. Epigenetics* **9**, 1–20 (2017).
14. Porter, H. *et al.* Cohesin-independent STAG proteins interact with RNA and R-loops and promote complex loading. *Elife* **12**, 1–33 (2023).
15. Nolen, L. D., Boyle, S., Ansari, M., Pritchard, E. & Bickmore, W. A. Regional chromatin decompaction in cornelia de lange syndrome associated with NIPBL disruption can be uncoupled from cohesin and CTCF. *Hum. Mol. Genet.* **22**, 4180–4193 (2013).
16. Davidson, I. F. *et al.* DNA loop extrusion by human cohesin. *Science (80-.)*. **366**, 1338–1345 (2019).
17. Petela, N. J. *et al.* Scc2 Is a Potent Activator of Cohesin's ATPase that Promotes Loading by Binding Scc1 without Pds5. *Mol. Cell* **70**, 1134–1148.e7 (2018).
18. Kim, Y., Shi, Z., Zhang, H., Finkelstein, I. J. & Yu, H. Human cohesin compacts DNA by loop extrusion. *Science (80-.)*. **366**, 1345–1349 (2019).

19. Rhodes, J., Mazza, D., Nasmyth, K. & Uphoff, S. Scc2/Nipbl hops between chromosomal cohesin rings after loading. *Elife* **6**, 1–20 (2017).
20. Wike, C. L. *et al.* Chromatin architecture transitions from zebrafish sperm through early embryogenesis. *Genome Res.* **31**, 981–994 (2021).
21. Horsfield, J. A. *et al.* Cohesin-dependent regulation of Runx genes. *Development* **134**, 2639–2649 (2007).
22. Zhang, S., Übelmesser, N., Barbieri, M. & Papantonis, A. Enhancer–promoter contact formation requires RNAPII and antagonizes loop extrusion. *Nat. Genet.* **55**, 832–840 (2023).
23. Barshad, G. *et al.* RNA polymerase II dynamics shape enhancer-promoter interactions. *Nat. Genet.* (2023) doi:10.1038/s41588-023-01442-7.
24. Dequeker, B. J. H. *et al.* MCM complexes are barriers that restrict cohesin-mediated loop extrusion. *Nature* **606**, 197–203 (2022).
25. Rao, S. S. P. *et al.* Cohesin Loss Eliminates All Loop Domains. *Cell* **171**, 305–320.e24 (2017).
26. Polovnikov, K. E. *et al.* Crumpled Polymer with Loops Recapitulates Key Features of Chromosome Organization. *Phys. Rev. X* **13**, 41029 (2023).
27. Holzmann, J. *et al.* Absolute quantification of cohesin, CTCF and their regulators in human cells. *Elife* **8**, 1–31 (2019).
28. Cattoglio, C. *et al.* Determining cellular CTCF and cohesin abundances to constrain 3D genome models. *Elife* **8**, (2019).
29. Kimmel, C. B., Ballard, W. W., Kimmel, S. R., Ullmann, B. & Schilling, T. F. Stages of embryonic development of the zebrafish. *Dev. Dyn.* **203**, 253–310 (1995).
30. Siefert, J. C. *et al.* Cell Cycle Control in the Early Embryonic Development of Aquatic Animal Species HHS Public Access Author manuscript. *Comp Biochem Physiol C Toxicol Pharmacol* **178**, 8–15 (2015).
31. Zhang, N. *et al.* A handcuff model for the cohesin complex. *J. Cell Biol.* **183**, 1019–1031 (2008).
32. Zhang, N. *et al.* Characterization of the Interaction between the Cohesin Subunits Rad21 and SA1/2. *PLoS One* **8**, (2013).
33. Cheng, H., Zhang, N. & Pati, D. Cohesin subunit RAD21: From biology to disease. *Gene* **758**, 144966 (2020).
34. Kane, D. A. *et al.* The zebrafish epiboly mutants. *Development* **123**, 47–55 (1996).
35. Mishima, Y. & Tomari, Y. Codon Usage and 3' UTR Length Determine Maternal mRNA Stability in Zebrafish. *Mol. Cell* **61**, 874–885 (2016).
36. Pálffy, M., Schulze, G., Valen, E. & Vastenhouw, N. L. Chromatin accessibility established by Pou5f3, Sox19b and Nanog primes genes for activity during zebrafish genome activation. *PLoS Genet.* **16**, 1–25 (2020).
37. Lee, M. T. *et al.* Nanog, Pou5f1 and SoxB1 activate zygotic gene expression during the maternal-to-zygotic transition. *Nature* **503**, 360–364 (2013).
38. Vliegthart, A. D. B. *et al.* Characterization of triptolide-induced hepatotoxicity by imaging and transcriptomics in a novel zebrafish model. *Toxicol. Sci.* **159**, 380–391 (2017).

39. Song, Y. S. *et al.* Validation, Optimization, and Application of the Zebrafish Developmental Toxicity Assay for Pharmaceuticals Under the ICH S5(R3) Guideline. *Front. Cell Dev. Biol.* **9**, (2021).
40. Huo, J. *et al.* Triptolide-induced hepatotoxicity via apoptosis and autophagy in zebrafish. *J. Appl. Toxicol.* **39**, 1532–1540 (2019).
41. Fudenberg, G., Abdennur, N., Imakaev, M., Goloborodko, A. & Mirny, L. A. Emerging Evidence of Chromosome Folding by Loop Extrusion. *Cold Spring Harb. Symp. Quant. Biol.* **82**, 45–55 (2017).
42. Rhodes, J. D. P. *et al.* Cohesin Can Remain Associated with Chromosomes during DNA Replication. *Cell Rep.* **20**, 2749–2755 (2017).
43. Mazza, D., Abernathy, A., Golob, N., Morisaki, T. & McNally, J. G. A benchmark for chromatin binding measurements in live cells. *Nucleic Acids Res.* **40**, 1–13 (2012).
44. Shaban, H. A., Barth, R., Recoules, L. & Bystricky, K. Hi-D: Nanoscale mapping of nuclear dynamics in single living cells. *Genome Biol.* **21**, 1–21 (2020).
45. Veith, R. *et al.* Balbiani ring mRNPs diffuse through and bind to clusters of large intranuclear molecular structures. *Biophys. J.* **99**, 2676–2685 (2010).
46. Höfling, F. & Franosch, T. Anomalous transport in the crowded world of biological cells. *Reports Prog. Phys.* **76**, 046602 (2013).
47. Woringer, M. & Darzacq, X. Protein motion in the nucleus: from anomalous diffusion to weak interactions. *Biochem. Soc. Trans.* **46**, 945–956 (2018).
48. Gebhardt, J. C. M. *et al.* Single-molecule imaging of transcription factor binding to DNA in live mammalian cells. *Nat. Methods* **10**, 421–426 (2013).
49. Speil, J. *et al.* Activated STAT1 transcription factors conduct distinct saltatory movements in the cell nucleus. *Biophys. J.* **101**, 2592–2600 (2011).
50. Izeddin, I. *et al.* Single-molecule tracking in live cells reveals distinct target-search strategies of transcription factors in the nucleus. *Elife* **2014**, (2014).

Response to reviewers' comments

Reviewer #1 (Remarks to the Author):

I appreciated the authors' detailed response to my comments and those of the other reviewers. I think it was a good call to leave out the Wapl experiments despite their promise, owing to the substantial extra characterisation needed in a first-time knock down model. I agree that the omission of these results does not impact on the overall findings. I'm comfortable that the revised manuscript is robust, and is of significance to the field.

We thank the reviewer for appreciating our work.

Minor comment:

- could the authors please label the Stag subunit in figure 1a? Confusing if not labelled, one might assume it's a structure like the ATPase heads of the SMC proteins.

As suggested, we adjusted figure 1a to include Stag1/2 labeling.

Reviewer #2 (Remarks to the Author):

I thank the authors for successfully addressing all my concerns by new experimental data and clarifications. In particular, the subcellular fractionation experiments demonstrating chromatin binding of the HT-Rad21 and HT-CTCF and the loss of this binding in mutant alleles HT-Rad21-3x and HT-CTCF- Δ ZF4-7 are valuable controls to support their conclusions. This also applies to the Co-IP experiments showing the formation of cohesin complexes by HT-Rad21, and to the phenotypic characterization of injected embryos.

I think that this work is an important contribution to the field and that this revised version is very suitable for publication in Nature Communications.

We thank the reviewer for appreciating our work.

Reviewer #3 (Remarks to the Author):

Cossmann et al. have resubmitted a revised manuscript, and the reviewer response is quite extensive and it is clear the authors have put a lot of work and care into the revisions.

Generally speaking, I found the revisions well-done. There were also a couple of things I missed (seems like I used the wrong URL to access the data due to a line number issue and a couple of things I asked for was already in SI Figures).

As I mentioned in the first review, I generally find the study compelling and innovative, but my major concern is the extremely low bound fraction of cohesin which appears incompatible with the requirement for cohesin in replication. The authors to their credit explicitly discuss this in their Discussion section – and I think including this is absolutely essential. I do find it much more likely that

the authors slightly misestimated the bound fraction than I do there being >1million cohesins per nucleus.

Along these lines the authors write “The number of cohesive cohesin molecules required for sister chromatid cohesin is not expected to change during development.”. Unless “per cell” is included, this statement is incorrect. If the maternally deposited number of cohesin proteins complexes is M and the number of cohesins required per nucleus is N and the cell division number is X, then the number of cohesins required is N^X which will grow rapidly and quickly exceed M. Every time the number of nuclei double during embryogenesis, the required number of cohesins per embryo also double. So, the authors should modify this sentence.

But at the end of the day, if the authors feel comfortable with this conclusion I think the study should get published as long as the critical discussion they included in the revision is also included in the final paper, then readers can decide for themselves.

The critical discussions are included in the final paper as they were sent to the reviewers. We only added the suggested “per cell” statement, as this clarification accurately reflects our intended meaning and we thank the reviewer for noticing this aspect.

I also think it would be good for the authors to state that they cannot exclude a subpopulation of very long-lived cohesins missed by the single-molecule tracking, since cohesive cohesins will have lifetimes of at least S- + G2-phase duration, which appears to exceed their currently estimated residence time.

We had already included such a statement in our discussion: “However, due to experimental limitations such as photobleaching and movement of nuclei in the living embryo, we cannot exclude that a fraction of molecules stay bound for longer times. The longest average residence times we obtained from GRID analysis should represent a lower limit.”

Thus, in conclusion I think this is a technically very impressive study that addresses some very interesting and previously unaddressed questions related to the establishment of 3D chromatin architecture in embryogenesis. I do have concerns about the low bound fractions and short residence times of cohesins, but as long as these are openly and explicitly discussed as the authors do in their current revision, I consider all my questions and concerns addressed and recommend the paper for publication in Nat Comm.

We thank the reviewer for the careful consideration and final positive evaluation of our manuscript. As mentioned above, we added all requested changes and the critical points are openly and explicitly discussed.

VERY SMALL THINGS:

Line 166 where they cite 101-103 REFs, this was also shown in PMID: 35420890 which should also be cited here.

We added the provided citation.